# Linear and Nonlinear Stability Analyses of Double-Diffusive Convection in a Vertical Brinkman Porous Enclosure under Soret and Dufour Effects

**Amel Bouachir** [1,*], **Mahmoud Mamou** [2], **Redha Rebhi** [3,4] and **Smail Benissaad** [1]

1 Laboratoire d'Énergétique Appliquée et de Pollution, Département de Génie Mécanique, Université Frères Mentouri Constantine 1, 325 Route Ain El Bey, Constantine 25017, Algeria; benissaad.smail@gmail.com
2 Aerospace Research Centre, Aerodynamics Laboratory, National Research Council, Ottawa, ON K1A 0R6, Canada; mahmoud.mamou@nrc.ca
3 Department of Mechanical Engineering, Faculty of Technology, University of Medea, Medea 26000, Algeria; rebhi.redha@gmail.com
4 LERM—Renewable Energy and Materials Laboratory, University of Medea, Medea 26000, Algeria
* Correspondence: bouachiramel26@gmail.com

**Abstract:** Analytical and numerical investigations were performed to study the influence of the Soret and Dufour effects on double-diffusive convection in a vertical porous layer filled with a binary mixture and subject to horizontal thermal and solute gradients. In particular, the study was focused on the effect of Soret and Dufour diffusion on bifurcation types from the rest state toward steady convective state, and then toward oscillatory convective state. The Brinkman-extended Darcy model and the Boussinesq approximation were employed to model the convective flow within the porous layer. Following past laboratory experiments, the investigations dealt with the particular situation where the solutal and thermal buoyancy forces were equal but acting in opposite direction to favor the possible occurrence of the rest state condition. For this situation, the onset of convection could be either supercritical or subcritical and occurred at given thresholds and following various bifurcation routes. The analytical investigation was based on the parallel flow approximation, which was valid only for a tall porous layer. A numerical linear stability analysis of the diffusive and convective states was performed on the basis of the finite element method. The thresholds of supercritical, $R_{TC}^{sup}$, and overstable, $R_{TC}^{over}$, convection were computed. In addition, the stability of the established convective flow, predicted by the parallel flow approximation, was studied numerically to predict the onset of Hopf's bifurcation, $R_{TC}^{Hopf}$, which marked the transition point from steady toward unsteady convective flows; a route towards the chaos. To support the analytical analyses of the convective flows and the numerical stability methodology and results, nonlinear numerical solutions of the full governing equations were obtained using a second-order finite difference method. Overall, the Soret and Dufour effects were seen to affect significantly the thresholds of stationary, overstable and oscillatory convection. The Hopf bifurcation was marked by secondary convective flows consisting of superposed vertical layers of opposite traveling waves. A good agreement was found between the predictions of the parallel flow approximation, the numerical solution and the linear stability results.

**Keywords:** Soret; Dufour; double diffusive; porous medium; subcritical convection; supercritical convection; Hopf bifurcation

## 1. Introduction

In recent years, combined thermo-diffusion and diffusion-thermal in double-diffusive convection occurring in fluid mixtures within saturated porous media had attracted many researchers' attention, owing to its importance in many applications such as in hydrology, petrology, geosciences, moisture transport, nuclear waste disposals, and solar ponds. The convection phenomenon is basically the result of the coexistence of temperature and

concentration gradients and the species coupling diffusion effects in fluids or in fluid-saturated porous media, under gravity effect. The coupling effect was described physically by the induction of a solute transfer caused by a temperature gradient, known as the Soret effect (thermo-diffusion), and also, a heat transfer caused by a concentration gradient, the so-called Dufour effect (diffusion-thermal), as reported by Nield and Bejan [1]. Most of the past investigations on double-diffusive convection carried out so far ignored this coupling effect to some extent, especially the Dufour effect that was believed to be negligible compared with heat and mass fluxes quantified by the Fourier's and Fick's laws. This assumption could be true; in fact, it is well known that Dufour effect is weak in liquids but significant in some gas mixtures and cannot be neglected as reported by Platten and Legros [2]. In the classical experiments on double-diffusive convection carried out by Krishnamurti [3,4] with the aim to disregard the Soret and Dufour effects, a removable barrier was used to separate the salt-stratified fluid from a sugar-stratified fluid. The removal of the barrier created a continuous horizontal gradient of sugar in the fluid, and the density was compensated by opposing horizontal gradients of salt, which is different from the discontinuity of properties in past experiments. First and foremost, the Dufour effect was discovered in gases by Clusius and Waldman [5], then it was firstly studied in the laboratory by Waldman [6]. Later on, many experimental studies were carried out which focused on the Dufour effect in fluids and binary fluids mixtures [7–10]. The principal purpose was to measure the value of the Dufour effect in the systems considered. The obtained results indicated that, the Dufour effect had a very small value when the initial concentration gradient imposed within the system decreased.

The role of the Dufour effect on the Raleigh–Bénard convection in binary gas mixtures was investigated theoretically by Hort et al. [11] and experimentally by Liu and Ahlers [12]. The Dufour effect had a significant influence on the topology and on the stability properties in liquid mixtures, while its influence was slightly less in real gas mixtures. The Soret effect was discovered by Ludwig [13] and studied later in detail by Charles Soret [14], where it was demonstrated that a salt solution contained in a tube with the two ends at different temperatures did not remain uniform in composition and a salt flux was generated by a temperature gradient under steady-state condition [15,16]. Platten [17] presented different techniques used to measure the Soret coefficient. Weaver and Viskanta [18] studied the influence of species inter-diffusion; Soret and Dufour effects, on the natural convection due to horizontal temperature and concentration gradients in a cavity. A recent comprehensive review of the natural convection due to combined thermal and solute driving forces was conducted by Nield and Bejan [1], Ingham and Pop [19] and Vafai [20].

Most of the past studies on double diffusive convection were concerned with vertical rectangular cavities for which the total buoyancy forces generated in the binary mixture were induced by the imposition of both thermal and solute gradients in the systems with negligible Soret and Dufour effects. For the particular situation where the thermal and solute buoyancy forces were opposing each other and of equal intensity, Trevisan and Bejan [21] developed an analytical solution valid only when the Lewis number is equal to unity, where the rest state was believed to be the only possible solution. Mamou et al. [22] analyzed the stability of double diffusive convection in a vertical rectangular porous enclosure. On the basis of the linear stability theory, the thresholds for the onsets of supercritical, oscillating, and overstability convection were determined. A threshold for the onset of subcritical finite-amplitude convection was computed analytically as function of the Lewis number; see also Mamou and Vasseur [23]. A stability analysis of the pure diffusive state and fully developed flows within a vertical porous layer was conducted by Mamou [24]. The same problem was reconsidered by Mamou et al. [25] and Karimi-Fard et al. [26] for an inclined rectangular cavity. The Brinkman-extended Darcy's law was employed by Amahmid et al. [27] to investigate the thermosolutal natural convection in a vertical porous layer. A linear stability analysis of double diffusive convection in a vertical Brinkman porous enclosure was performed by Mamou et al. [28]. Furthermore, a

three-dimensional doubly diffusive convection in a binary fluid was studied by Beaume et al. [29,30] and Bergeon and Knobloch [31].

Other investigations concerning thermal-diffusion or Soret-induced convection related to the current subject were carried out in vertical fluid and porous cavities. In these problems, both thermal and solutal buoyancy forces in the binary mixture were the consequence of the imposition of a temperature gradient only across the system. The condition of opposing and equal thermal and solutal buoyancy forces was the focus in these investigations. For this condition, Marcoux et al. [32] investigated numerically the onset of thermo-gravitational diffusion in a porous medium saturated by a binary mixture subject to the Soret effect. The separation in an inclined porous cell saturated by a binary mixture was investigated analytically and numerically, and also experimentally using a solution of CuSO4 as reported in El Hajjar et al. [33]. The Soret effect in a porous media system sandwiched between two layers of identical binary hydrocarbon mixture was studied numerically and experimentally by Ahadi et al. [34]. The Brinkman-extended Darcy model was used by Joly et al. [35,36] to analyze the Soret effect on the onset of convection in a vertical porous enclosure. Both double-diffusive and Soret-induced convection in a vertical porous layer were studied analytically and numerically by Boutana et al. [37]. The results showed that the flow patterns induced by both double-diffusive and Soret-induced convection were qualitatively similar but quantitatively different. Er-Raki et al. [38] studied the Soret effect on double-diffusive convection generated in a vertical porous layer. More recently, the Soret convection in a vertical porous enclosure under the influence of the form drag was analyzed by Rebhi et al. [39]. The binary fluid flow in the porous medium was described by the Darcy–Dupuit model. The thermodiffusion phenomenon in binary and ternary liquid mixtures was well documented by Köhler et al. [40]. Costesèque et al. [41] indicated the need for more laboratory studies on thermodiffusion and thermodiffusion-convection transport in porous media to accurately model the phenomenon and understand the behavior of multicomponent mixtures, and to measure and provide the appropriate effective values of thermodiffusion, diffusion, and cross-diffusion coefficients.

Recently, a few more studies regarding double-diffusive natural convection of binary fluids in porous enclosures were accomplished by taking the Soret and Dufour effects into account. Bella et al. [42,43] investigated the influence of the Soret and Dufour effects on double diffusive free convection and double diffusive magneto-hydrodynamic natural convection in an inclined square porous cavity. Dirichlet boundary conditions for temperature and solute were imposed on the two active walls, while the two other walls were impermeable and adiabatic. Motsa [44] studied the influence of Soret and Dufour effects on the onset of convection using a linear stability analysis. The result demonstrated that the Soret and Dufour coefficients had stabilizing and destabilizing effects, respectively, on stationary instability, while they didn't have an effect on the onset of overstability. Soret and Dufour effects on unsteady double diffusive convection in a square cavity filled with a gaseous binary mixture were examined by Ben Niche et al. [45]. Nithyadevi and Yang [46] numerically analyzed the effects of various governing parameters on water convective flow and on heat and mass transfer rates in a partially heated square cavity in the presence of Soret and Dufour effects. The transient double-diffusive convection in a rectangular vertical layer, subjected to constant and different temperatures and concentrations on vertical walls, was analyzed numerically by Ren and Chan [47]. The influence of the governing parameters on the resulting fluid flow, temperature, and concentration fields was discussed in details. More recently, Lagra et al. [48], Attia et al. [49], and Hasnaoui et al. [50] investigated analytically and numerically the Soret and Dufour effects on thermosolutal convection induced in a horizontal layer subject to constant heat and mass fluxes. The influence of the Soret and Dufour effects on the thresholds of stationary convection, subcritical convection, flow structure, and heat and mass transfer rates were discussed. The thermosolutal natural convection, generated in an inclined square cavity filled with a binary fluid in the presence of the Soret and Dufour effects, was investigated numerically by Hasnaoui et al. [51] using a hybrid Lattice Boltzmann finite difference method.

Toward knowledge enrichment and physical understanding of the influence of Soret and Dufour effects on double-diffusive natural convection in vertical fluid or porous layers, double-diffusive convection in binary mixtures was considered in the present paper. The convective flow was modeled according to the unsteady Brinkman-extended Darcy law. Horizontal gradients of temperature and concentration were imposed on the vertical walls. The particular situation where the thermal and solutal buoyancy forces were equal and opposing each other was examined. The main objective of the present investigation was to analyze the influence of the Soret and Dufour effects on the flow structure and on the heat and mass transfer rates. The investigation is unique and somehow fairly complete as it is based on various corroborated approaches, which were based on numerical, asymptotic, and linear and nonlinear stability analyses. The parallel flow approximation was used to find the threshold of the subcritical convection, which was characterized by the critical Rayleigh number. On the basis of the finite element method, a linear stability analysis was performed to predict the thresholds of supercritical, overstable, and oscillatory convection. A linear stability analysis was conducted as well to find the onset of Hopf bifurcation. The combined effects of Soret and Dufour and other governing parameters on the induced convective flows were discussed and analyzed, and they were presented in terms of the stream function, temperature, and concentration profiles and heat and solute transfer rates, and stability diagrams.

## 2. Problem Description and Mathematical Formulation

The current investigation is on double-diffusive convection instability in a vertical rectangular porous enclosure having an aspect ratio of $A = H'/L'$, where $H'$ is the height and $L'$ is the width of the enclosure. The origin of the coordinate system is situated in the center of the cavity with $x'$ and $y'$ representing, respectively, the horizontal axis pointing to the right and the vertical axis pointing upward, as illustrated in Figure 1. Neumann boundary conditions for both temperature and concentration $q'$ and $j'$ are applied on the vertical walls of the enclosure. The short horizontal walls of the cavity are considered adiabatic and impermeable. The porous medium is assumed to be isotropic, homogeneous, and saturated by a Newtonian and incompressible binary mixture, where the Dufour and Soret effects are considered. Using the Boussinesq approximation, the density variation $\rho$ with temperature $T'$ and concentration $S'$ is described by the linearized state equation as $\rho = \rho_0[1 - \beta_T(T' - T'_0) - \beta_S(S' - S'_0)]$, where $\beta_T$ and $\beta_S$ are the thermal and concentration expansion coefficients, respectively, and they are defined as:

$$\beta_T = -\frac{1}{\rho_0}\left(\frac{\partial \rho}{\partial T'}\right)_{P',S'}, \quad \beta_S = -\frac{1}{\rho_0}\left(\frac{\partial \rho}{\partial S'}\right)_{P',\, T'}$$

where $\rho_0$ is the fluid mixture density at temperature $T'_0$ and concentration $S'_0$, which represents the temperature and concentration values at the origin of the coordinate system.

The heat flux induced by conduction and diffusion-thermal (Dufour effect) is expressed as [1]:

$$\overrightarrow{q'} = -k_p\,\nabla T' - D_{TS}\,\nabla S'$$

where $k_P$ and $D_{TS}$ are the thermal conductivity of the saturated porous medium and the Dufour coefficient, respectively.

The Fick's law of mass diffusion [1], in the presence of thermo-diffusion (Soret effect) induced by the imposition of a temperature gradient, is defined as follows:

$$\overrightarrow{j'} = -D\,\nabla S' - D_{ST}\,\nabla T'$$

where $D$ and $D_{ST}$ are respectively the mass diffusivity of saturated porous medium and the thermal-diffusion coefficient.

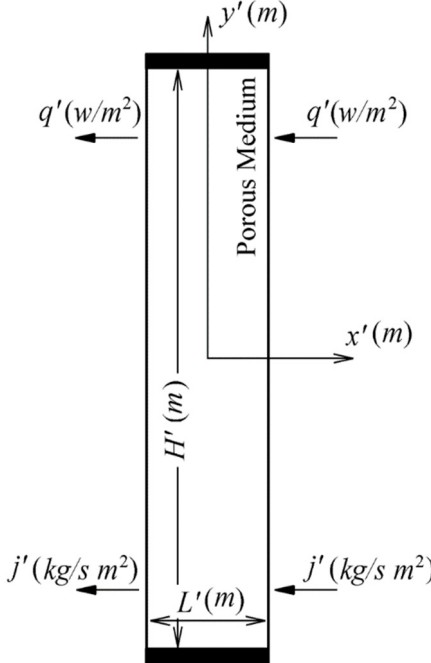

**Figure 1.** The physical model and coordinate system.

Adopting unsteady Brinkman-extended Darcy's model [1] and taking into account the Soret and Dufour effects [1], the governing equations expressing conservation of mass, momentum, energy, and species are given as follows:

$$\frac{\partial u'}{\partial x'} + \frac{\partial v'}{\partial y'} = 0 \tag{1}$$

$$\frac{K}{\varepsilon' v}\frac{\partial u'}{\partial t'} + u' = -\frac{K}{\mu}\left[\frac{\partial P'}{\partial x'} - \mu_e\left(\frac{\partial^2 u'}{\partial x'^2} + \frac{\partial^2 u'}{\partial y'^2}\right)\right] \tag{2}$$

$$\frac{K}{\varepsilon' v}\frac{\partial v'}{\partial t'} + v' = -\frac{K}{\mu}\left[\frac{\partial P'}{\partial y'} - \mu_e\left(\frac{\partial^2 v'}{\partial x'^2} + \frac{\partial^2 v'}{\partial y'^2}\right) + \rho_0 g\left[\beta_T\left(T' - T_0'\right) + \beta_S\left(S' - S_0'\right)\right]\right] \tag{3}$$

$$\sigma\frac{\partial T'}{\partial t'} + \overrightarrow{V'}\,\nabla T' = \alpha\,\nabla^2 T' + D_{TS}\nabla^2 S' \tag{4}$$

$$\varepsilon'\frac{\partial S'}{\partial t'} + \overrightarrow{V'}\,\nabla S' = D\,\nabla^2 S' + D_{ST}\nabla^2 T' \tag{5}$$

where $\overrightarrow{V'}$ is the velocity vector, $u'$ and $v'$ represent the velocity components, $t'$ the time, $P'$ the pressure, $K$ the porous medium permeability, $\varepsilon'$ the porous medium porosity, $v$ the kinematic viscosity of the fluid $(v = \mu/\rho)$, $\mu$ the dynamic viscosity of the fluid, $\mu_e$ the effective dynamic viscosity, $g$ the gravitational acceleration, $\sigma$ the saturated porous medium to fluid heat capacities ratio $\left(\sigma = (\rho c)_p/(\rho c)_f\right)$, and $\alpha$ the thermal diffusivity of the saturated porous medium $\left(\alpha = k_p/(\rho c)_f\right)$.

For a parametric study, the following dimensionless variables are used to put the above Equations (1)–(5) into a dimensionless form:

$$(x, y) = \left(\frac{x'}{L'}, \frac{y'}{L'}\right), \ (u, v) = \left(\frac{u'}{U^*}, \frac{v'}{U^*}\right), \ t = \frac{t'}{t^*}, \ T = \frac{(T' - T_0')}{\Delta T^*},$$

$$S = \frac{(S' - S_0')}{\Delta S^*}, \ P = \frac{P'}{P^*}$$

where $t^*$, $U^*$, $P^*$, $\Delta T^*$, and $\Delta S^*$ are the characteristic time, velocity, pressure, temperature and concentration scales defined as:

$$U^* = \frac{\alpha}{L'}, \quad t^* = \sigma \frac{L'}{U^*}, \quad P^* = \frac{\rho_0 \, Pr U^{*2}}{Da}, \quad \Delta T^* = \frac{q'L'}{k_p}, \quad \Delta S^* = \frac{j'L'}{D}$$

where $Pr = v/\alpha$ is the Prandtl number and $Da = K/L'^2$ is the Darcy number.

Using the stream-function formulation, the stream-function $\Psi$ is linked to the velocity components as $u = \partial\Psi/\partial y$ and $v = -\partial\Psi/\partial x$ such that the continuity equation is satisfied. Eliminating the pressure from Equations (2) and (3), the governing Equations (1)–(5) in terms of $\Psi$ are rewritten in a dimensionless form as follows:

$$\xi \frac{\partial \left(\nabla^2 \Psi\right)}{\partial t} + \nabla^2 \Psi = Da_e \Delta^2 \Psi - R_T \left( \frac{\partial T}{\partial x} + N \frac{\partial S}{\partial x} \right) \tag{6}$$

$$\frac{\partial T}{\partial t} + \frac{\partial \Psi}{\partial y}\frac{\partial T}{\partial x} - \frac{\partial \Psi}{\partial x}\frac{\partial T}{\partial y} = \nabla^2 T + D_u \nabla^2 S \tag{7}$$

$$\varepsilon \frac{\partial S}{\partial t} + \frac{\partial \Psi}{\partial y}\frac{\partial S}{\partial x} - \frac{\partial \Psi}{\partial x}\frac{\partial S}{\partial y} = Le^{-1}\left( \nabla^2 S + S_r \nabla^2 T \right) \tag{8}$$

According to Equations (6)–(8), the present problem is governed by nine dimensionless parameters, which are the Rayleigh number, $R_T$, the effective Darcy number (called Darcy number hereafter), $Da_e$, the buoyancy ratio, $N$, the Lewis number, $Le$, the Soret parameter, $S_r$, the Dufour parameter, $D_u$, the aspect ratio of the cavity, $A$, the porous medium acceleration coefficient, $\xi$, and the normalized porosity of the porous medium, $\varepsilon$. They are expressed by:

$$\left. \begin{array}{c} R_T = \frac{\rho g \, \beta_T \, \Delta T^* \, L'^3}{\alpha \mu}, \quad Da_e = r_\mu Da \quad N = \frac{\beta_S \, \Delta S^*}{\beta_T \, \Delta T^*}, \quad Le = \frac{\alpha}{D}, \\[2mm] S_r = \frac{D_{ST} \, \Delta T^*}{D \, \Delta S^*}, \quad D_u = \frac{D_{TS} \, \Delta S^*}{\alpha \, \Delta T^*}, \quad A = \frac{H'}{L'}, \quad \xi = \frac{Da}{\varepsilon' \, \sigma Pr}, \quad \varepsilon = \frac{\varepsilon'}{\sigma} \end{array} \right\} \tag{9}$$

where the parameter $r_\mu = \mu_e/\mu$ is the effective viscosity to fluid viscosity ratio. It was commonly considered in the past studies as: $r_\mu = 1/\varepsilon'$ or $r_\mu = 1$. For a Brinkman porous medium, $\varepsilon' \approx 1$, the viscosity ratio became close to unity ($\mu_e = \mu$) [1]. Furthermore, Givler and Altobelli [52] performed an experimental study for the determination of the effective viscosity for the Brinkman–Forchheimer flow model and found that the parameter $r_\mu \geq 1$.

The experimental values of the Soret and Dufour parameters ($S_r$ and $D_u$) are now discussed and an explanation on how their values vary is provided. On the one hand, the Soret number is defined as $S_T = D_{ST}/D$, where $D_{ST}$ is the physical thermodiffusion coefficient. For some typical fluid mixtures, such as: water-isopropanol mixtures (water mass fraction of 0.2), water-isopropanol mixture (water mass fraction of 0.9), water-methanol (water mass fraction of 0.1), and water-ethanol (water mass fraction of 0.1) the physical thermodiffusion number, $S_T$, was measured as $3.09 \times 10^{-3}\mathrm{K}^{-1}$ [53], $-8.47 \times 10^{-3}\mathrm{K}^{-1}$ [53], $1.88 \times 10^{-3}\mathrm{K}^{-1}$ [54], and $2.71 \times 10^{-3}\mathrm{K}^{-1}$ [54], respectively. Thus, the present Soret parameter $S_r$ becomes a function of the scaling factors as $S_r = S_T \Delta T^*/\Delta S^*$, which could make $S_r$ vary from 0 to plus or minus large values according to the values of the temperature and concentration difference and the sign of the Soret number, $S_T$. On the other hand, in the present notation and according to [55], the contribution to the net heat flux is the heat transfer by conduction and by Dufour diffusion, where the net heat flux is defined by:

$$-\frac{\vec{q'}}{\rho C_P} = \alpha \, \nabla T' + \rho \frac{\partial \mu_{S'}}{\partial S'} \, T_0' \, \frac{D''}{\rho C_P} \nabla S' \tag{10}$$

where $\mu_{S'}$ is the chemical potential and $D''$ is the Dufour coefficient. From this net heat flux relationship, the Dufour parameter $D_u$ is related to the Dufour coefficient as follows [12,55]:

$$D_u = Q \frac{\beta_S^2}{\beta_T^2} \frac{\Delta S^*}{\Delta T^*} \frac{1}{Le} \frac{D''}{D} \text{ with } Q = \frac{\partial \mu_{S'}}{\partial S'} \frac{T_0'}{C_P} \frac{\beta_T^2}{\beta_S^2} \tag{11}$$

In the 1960s, it was demonstrated that the Dufour effect does exist in liquid mixtures [8,56], however, it was discovered in gases a few decades earlier [57]. Some experimental values in liquids mixtures were reported for a large number of mixtures and the Dufour coefficient $D''$ was significant. In gases mixtures, Liu and Ahlers [12] reported some typical values of the Dufour effect which were given in terms of Dufour number $Q$, as stated in the relationship above, which is a purely thermodynamic quantity. The Soret coefficient could be negative or positive, however the Dufour number is always positive. As discussed in Liu and Ahlers [12], the quantity $Q$ could be very large, $Q \sim O (1)$–$O (2)$. The Lewis number is of order $O (1)$ for gases and $O (2)$ for liquids. Other parameters can be computed for any fluid or gas mixtures and it can be shown that the parameter $D_u$ varies from 0 to large positive values. Liu and Ahlers [12] reported a large number of gases mixtures with Soret and Dufour coefficient measurements. In the present investigation, we focus on a parametric study, which could suit any working fluid mixtures, and the parameters in the Sr and Du expression above could be tuned up to obtain the desired Soret and Dufour parameters.

The boundary conditions imposed on the system are expressed in a dimensionless form as:

$$x = \pm \frac{1}{2}: \quad \Psi = \frac{\partial \Psi}{\partial x} = 0, \quad \frac{\partial T}{\partial x} + D_u \frac{\partial S}{\partial x} = 1, \quad \frac{\partial S}{\partial x} + S_r \frac{\partial T}{\partial x} = 1 \tag{12}$$

for the active vertical walls, and they can be reduced to:

$$\frac{\partial T}{\partial x} = a_T \text{ and } \frac{\partial S}{\partial x} = a_S$$

where $a_T$ and $a_S$ are defined as:

$$a_T = \frac{1 - D_u}{1 - D_u S_r}, \quad a_S = \frac{1 - S_r}{1 - D_u S_r} \tag{13}$$

and:

$$y = \pm \frac{A}{2}: \quad \Psi = \frac{\partial \Psi}{\partial y} = 0, \quad \frac{\partial T}{\partial y} = \frac{\partial S}{\partial y} = 0 \tag{14}$$

for the adiabatic and impermeable walls.

In the present paper, the case where the resultant of the thermal and solutal buoyancy forces is nil is considered such that:

$$N = -\frac{a_T}{a_S} \tag{15}$$

For this particular buoyancy ratio value, the rest state is a possible solution of the problem, but it becomes unstable above a threshold and can bifurcate toward a convective state. In this study with the current problem formulation we assume that $D_u S_r \neq 1$ to avoid singular conditions.

The local and average heat and mass transfer rates expressed in terms of the Nusselt and Sherwood number are defined respectively as:

$$\left.\begin{array}{ll} Nu = \frac{1}{\Delta T + D_u \Delta S}, & Sh = \frac{1}{\Delta S + S_r \Delta T} \\ Nu_m = \frac{1}{A} \int_{-A/2}^{A/2} Nu \, dy, & Sh_m = \frac{1}{A} \int_{-A/2}^{A/2} Sh \, dy \end{array}\right\} \tag{16}$$

where $\Delta T = T(-1/2, y) - T(1/2, y)$ and $\Delta S = S(-1/2, y) - S(1/2, y)$ are the dimensionless temperature and concentration differences, respectively. The subscript $m$ denotes an average value along the vertical walls.

### 3. Numerical Solution

A finite difference method was used to solve numerically the full governing equations subject to the prescribed boundary conditions. The energy and concentration, Equations (7) and (8), together with the boundary conditions, Equations (12) and (14), were discretized using a second-order finite difference scheme in time and space with a uniform grid. The alternating direction implicit method (ADI) was employed for a time-accurate solution. The resulting sets of discretized equations for each variable were solved by a line-by-line procedure, using the tri-diagonal matrix algorithm (TDMA). However, the stream function equation, Equation (6), was solved using the successive over-relaxation method (SOR) with known temperature and concentration distributions from the previous time step. The temporal terms in all equations were discretized using a second-order backward difference scheme. The boundary conditions were also discretized using a second-order backward finite difference scheme. At each new time step the SOR iterative procedure was repeated until the following convergence criterion is reached:

$$\frac{\sum_i \sum_j \left| \Psi_{i,j}^{k+1} - \Psi_{i,j}^{k} \right|}{\sum_i \sum_j \left| \Psi_{i,j}^{k+1} \right|} \leq 10^{-6} \tag{17}$$

where $k$ denotes the $k$th iteration.

Depending on the governing parameters' values, owing to non-slip boundary conditions and thin viscous flow layers on the walls for small Darcy number, $Da_e$, the grid size of $200 \times 300$ was adopted for most of the cases considered in this study.

The accuracy of the present numerical solutions depends on the grid size $(N_x \times N_y)$. Thus, a grid sensitivity study is performed to find the adequate grid size beyond which the solutions become independent of the grid size. Various grid sizes are considered as shown in Table 1. The predicted numerical results are compared with the exact analytical solution valid for an infinite layer. For the numerical solution, an aspect ratio of A = 10 is considered, which nearly mimics the infinite layer flow. The numerical results are obtained for $Da_e = 1$, $R_T = 10^4$, $Le = 10$, and $D_u = S_r = 0.1$. According to Table 1, grid refinement is seen to improve the results accuracy as they become independent of the grid size beyond $100 \times 200$. Therefore, to be more conservative, a grid size of $200 \times 300$ is adopted and it is believed to provide the numerical solutions with sufficient accuracy.

**Table 1.** Grid sensitivity study for $A = 10$, $Da_e = 1$, $R_T = 10^4$, $Le = 10$, and $D_u = S_r = 0.1$.

| $N_x \times N_y$ | Numerical Solution | | | | Analytical Solution |
|---|---|---|---|---|---|
| | $40 \times 80$ | $80 \times 160$ | $100 \times 200$ | $200 \times 300$ | |
| $\Psi_0$ | 3.7171 | 3.7861 | 3.7877 | 3.7906 | 3.7911 |
| Error (%) | 1.97 | 0.13 | 0.09 | 0.01 | Reference |
| $Nu$ | 2.4561 | 2.4406 | 2.4362 | 2.4312 | 2.4187 |
| Error (%) | 1.53 | 0.90 | 0.72 | 0.51 | Reference |
| $Nu_m$ | 2.4057 | 2.3871 | 2.3824 | 2.3773 | . . . |
| $Sh$ | 3.1944 | 3.1887 | 3.1823 | 3.1730 | 3.1072 |
| Error (%) | 2.77 | 2.59 | 2.39 | 2.10 | Reference |
| $Sh_m$ | 3.4365 | 3.4108 | 3.3989 | 3.3828 | . . . |

For validation of the present numerical solutions with past study numerical results, Table 2 illustrates the numerical solution obtained for $A = 8$, $R_T = 150$, $N = -1$, $Le = 10$, and $D_u = 0$ and $S_r = -1$, in terms of the maximum stream function value, and the local

Nusselt and Sherwood number for low and high Darcy number values, as defined in Joly et al. [36]. The results are compared with those reported by Joly et al. [36] with a very good agreement.

**Table 2.** Comparison of the present computed values of $\Psi_{max}$, $Nu$, and $Sh$ with a past study of numerical results for $A = 8$, $R_T = 150$, $N = -1$, $Le = 10$, and $D_u = 0$ and $S_r = -1$.

| | $Da_e = 1$ | | | $Da_e = 0.001$ | | |
|---|---|---|---|---|---|---|
| | Joly et al. [36] | Present Study | Error (%) | Joly et al. [36] | Present Study | Error (%) |
| $\Psi_{max}$ | 0.32 | 0.33 | 3.08 | 3.16 | 3.16 | 0 |
| $Nu$ | 1.03 | 1.03 | 0 | 2.79 | 2.80 | 0.36 |
| $Sh$ | 2.58 | 2.63 | 1.92 | 4.89 | 4.92 | 0.63 |

Typical numerical results are presented in Figure 2a–e for $A = 10$, $Le = 10$, $\varepsilon = 1$ and various values of the effective Darcy and Rayleigh numbers and the Soret and Dufour parameters. In these graphs, streamlines, isotherms, and isoconcentrations are illustrated from left to right, respectively. Independently of the governing parameters, the results distinctly show that the flow in the core region of a tall cavity $(A \gg 1)$ is essentially parallel, while the temperature and concentration are linearly stratified in the $y$-direction. These observations, which were reported in the past by several authors [22,27,36] and confirmed numerically by the results sketched in Figure 2, are the foundations of the parallel flow assumption, which was considered in the present study.

The effect of the effective Darcy number is depicted in Figure 2c–e for $A = 10$, $Le = 10$, $D_u = S_r = 0.1$, and $\varepsilon = 1$. When the effective Darcy number is relatively small, the viscous effect (Brinkman term) is negligible and the enclosure walls behave like non-slip walls as the viscous boundary layer becomes very thin when $Da_e \to 0$. This effect is illustrated in Figure 2a–c by the streamlines clustered near the walls. As $Da_e$ increases, the influence of the boundary effect on the flow, temperature, and concentration fields becomes more significant, where the strength of the flow circulation is decreased and the streamlines become sparsely distributed near the walls, owing to the viscous effect appearance.

The Soret and Dufour effects are presented in Figure 2a–c, where contours of the stream function, temperature, and concentration corresponding to $S_r = D_u = 0.1$ are similar to those obtained when $S_r = D_u = 0$ (not presented here). From Figure 2a,b, when the Dufour (Soret) effect is equal to 0.8, the thermal (Solute) buoyancy effects are dominant, which slightly enhances (diminishes) the flow intensity, where the stream function value increases from $\Psi_0 = 3.45$ to $\Psi_0 = 3.55$. Figure 3a indicates that when the Dufour parameter increases up to 0.8, the isotherm lines become less distorted due to a decrease in the thermal gradient, which is apparent from Equation (7). However, the isoconcentration lines are more distorted due to the increase in the concentration gradient (see Equation (8)), which increases the mass transfer rate from $Sh = 4.89$ to $Sh = 5.28$. Furthermore, Figure 2c clearly indicates that the Soret effect has an inverse trend.

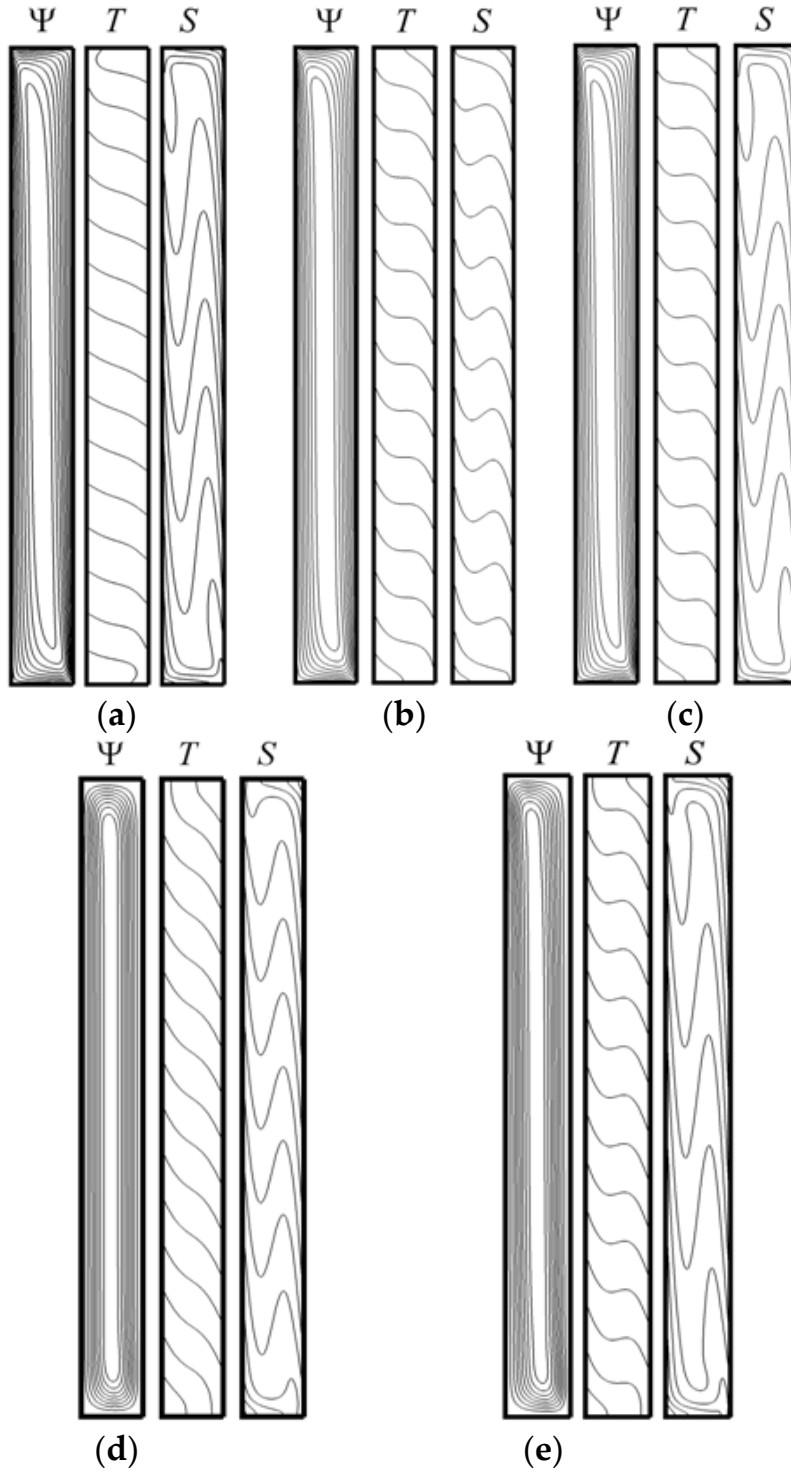

**Figure 2.** Contours of stream function (left), temperature (center), and concentration (right) for: $A = 10$, $Le = 10$, $\varepsilon = 1$: (**a**) $Da_e = 10^{-4}$, $R_T = 200$, $S_r = 0$, $D_u = 0.8$: $\Psi_0 = 3.55$, $Nu = 3.20$, and $Sh = 5.28$; (**b**) $Da_e = 10^{-4}$, $R_T = 200$, $S_r = 0.8$, $D_u = 0$: $\Psi_0 = 2.79$, $Nu = 2.74$, and $Sh = 2.88$; (**c**) $Da_e = 10^{-4}$, $R_T = 200$, $S_r = D_u = 0.1$: $\Psi_0 = 3.45$, $Nu = 3.22$, and $Sh = 4.89$; (**d**) $Da_e = 10^{-1}$, $R_T = 200$, $D_u = S_r = 0.1$: $\Psi_0 = 1.80$, $Nu = 1.64$ and $Sh = 2.91$; and (**e**) $Da_e = 10^{0}$, $R_T = 10^{4}$, $D_u = S_r = 0.1$: $\Psi_0 = 3.79$, $Nu = 2.43$, and $Sh = 3.17$.

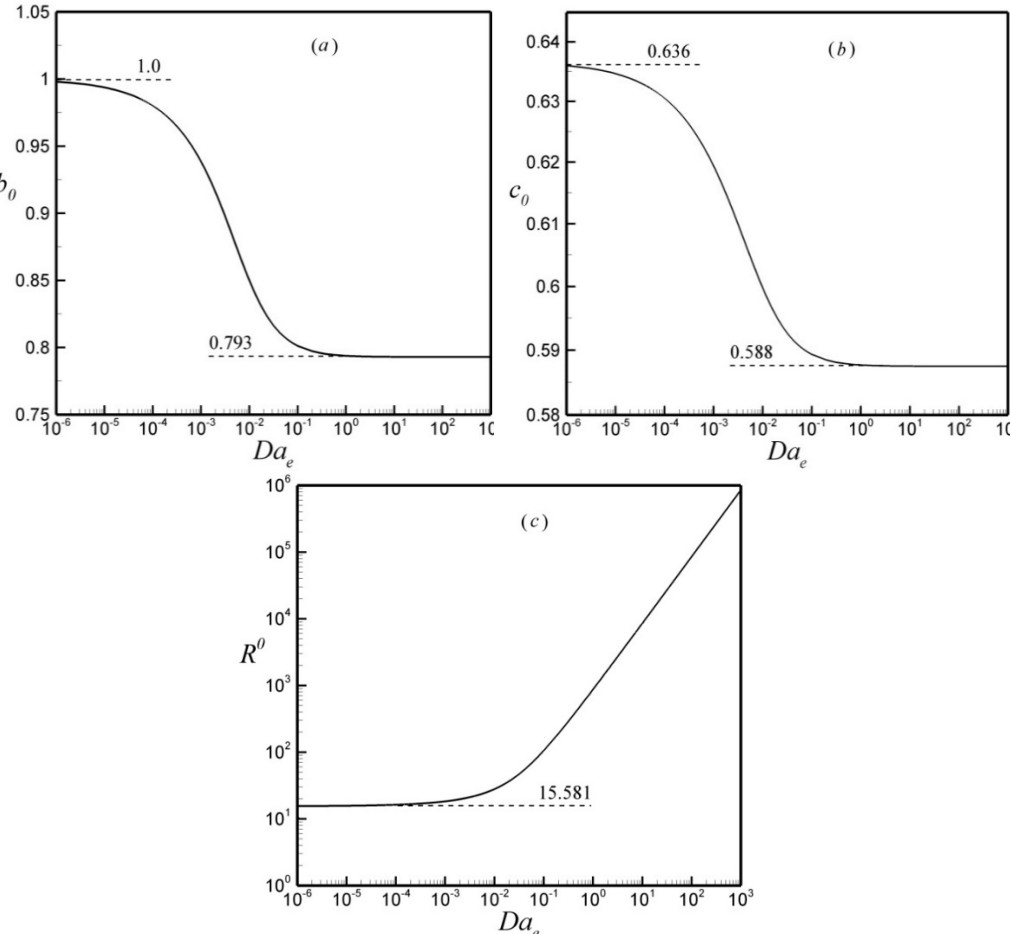

**Figure 3.** The effect of $Da_e$ on (**a**) $b_0$, (**b**) $c_0$, and (**c**) $R^0$ for an infinite layer $A \to \infty$.

## 4. Analytical Solution

For double-diffusion convection in a tall cavity having a large aspect ratio ($A >> 1$), several authors developed an approximate analytical solution based on the parallel flow approximation (see, for instance, Mamou et al. [22], Amahmid et al. [27], and Joly et al. [36]). For the present problem according to this approach and to the numerical observations made on Figure 2, the convective flow in the central region of the enclosure is assumed to be parallel to the vertical walls. Thus, the horizontal velocity component is neglected and the vertical velocity component depends now only on $x$. In this way, the stream function, temperature, and concentration can be approximated and written as follows:

$$\left.\begin{aligned}
\Psi(x,y) &\approx \Psi(x) \\
T(x,y) &\approx C_T y + \Theta_T(x) \\
S(x,y) &\approx C_S y + \Theta_S(x)
\end{aligned}\right\} \tag{18}$$

where $C_T$ and $C_S$ are, respectively, the unknown constant temperature and concentration gradients in the $y$-direction.

Introducing the above approximations, Equation (18), in the governing Equations (6)–(8) and making use of the boundary conditions (12)–(14), we obtain the following system of ordinary differential equations:

$$\frac{d^2\Psi}{dx^2} = Da_e \frac{d^4\Psi}{dx^4} - R_T \left( \frac{d\Theta_T}{dx} - a_T a_S^{-1} \frac{d\Theta_S}{dx} \right) \tag{19}$$

$$-C_T \frac{d\Psi}{dx} = \frac{d^2\Theta_T}{dx^2} + D_u \frac{d^2\Theta_S}{dx^2} \tag{20}$$

$$-C_S Le \frac{d\Psi}{dx} = \frac{d^2\Theta_S}{dx^2} + S_r \frac{d^2\Theta_T}{dx^2} \tag{21}$$

After performing a first integration of the energy and species Equations (20) and (21), and making use of the boundary conditions, Equation (12), it is readily found:

$$\frac{d\Theta_T}{dx} = \frac{C_S D_u Le - C_T}{1 - D_u S_r} \Psi + a_T \tag{22}$$

$$\frac{d\Theta_S}{dx} = \frac{C_T S_r - C_S Le}{1 - D_u S_r} \Psi + a_S \tag{23}$$

Substituting Equations (22) and (23) into the momentum Equation (19), we obtain the following ordinary simplified differential equation:

$$-Da_e \frac{d^4\Psi}{dx^4} + \frac{d^2\Psi}{dx^2} + \Omega^2 \Psi = 0 \tag{24}$$

where: $\Omega = [R_T(Le C_S - C_T)/(1 - S_r)]^{1/2}$.

The solution of Equation (24), satisfying the boundary conditions in Equation (12), and the stream function approximation stated in Equation (18), is obtained as follows:

$$\Psi(x) = \Psi_{0n} \frac{1}{\sqrt{\eta_1}} [\cos(\theta_2 x) - \eta_0 \cosh(\theta_1 x)] \tag{25}$$

where the constants $\theta_1$, $\theta_2$, $\eta_0$ and $\eta_1$ are defined by:

$$\left. \begin{array}{l} \theta_1 = \left( \frac{\sqrt{1 + 4Da_e \Omega^2} + 1}{2Da_e} \right)^{\frac{1}{2}}, \quad \theta_2 = \left( \frac{\sqrt{1 + 4Da_e \Omega^2} - 1}{2Da_e} \right)^{\frac{1}{2}}, \\[3mm] \eta_0 = \frac{\cos(\theta_2/2)}{\cosh(\theta_{1/2})}, \quad \eta_1 = 1 + \eta_0^2 + \frac{\sin(\theta_2)}{\theta_2} \left( 1 - \frac{\theta_2^2}{\theta_1^2} \right) \end{array} \right\} \tag{26}$$

and $\Psi_{0n}$ is the normalized stream function value at the central part of the cavity, and it is defined as follows:

$$\Psi_{0n} = \Psi_0 \sqrt{b_0} \tag{27}$$

where: $b_0 = \eta_1/(1 - \eta_0)^2$ and $\Psi_0 = \Psi(0)$.

In Equation (27), $\Psi_0$ is the stream function at the center of the cavity. From the boundary conditions for $\Psi$ and its derivative $\partial\Psi/\partial x$ at $x = \pm 1/2$, it is found that:

$$\theta_2 \tan\left(\frac{\theta_2}{2}\right) + \theta_1 \tanh\left(\frac{\theta_1}{2}\right) = 0 \tag{28}$$

The relationship between, $\theta_1$, $\theta_2$, and $\Omega$ is given by:

$$\theta_1^2 = \theta_2^2 + \frac{1}{Da_e}, \quad \Omega^2 = \theta_2^2 \left( Da_e \theta_2^2 + 1 \right) \tag{29}$$

Upon solving (22) and (23), one obtains the following solution:

$$T(x, y) = C_T y - \frac{1}{\sqrt{\eta_1}} \left( \frac{C_T - C_s D_u Le}{1 - D_u S_r} \right) \left( \frac{\sin(\theta_2 x)}{\theta_2} - \eta_0 \frac{\sinh(\theta_1 x)}{\theta_1} \right) \Psi_{0n} + a_T x \tag{30}$$

$$S(x, y) = C_S y - \frac{1}{\sqrt{\eta_1}} \left( \frac{C_S Le - C_T S_r}{1 - D_u S_r} \right) \left( \frac{\sin(\theta_2 x)}{\theta_2} - \eta_0 \frac{\sinh(\theta_1 x)}{\theta_1} \right) \Psi_{0n} + a_S x \tag{31}$$

In the past, Trevisan and Bejan [21] demonstrated that the parallel flow approximation is only applicable in the central region of the enclosure, but in the end regions where the

flow is more complicated, the boundary conditions in the *y*-direction, Equation (14), cannot be applied exactly with this approximation. For this reason, these conditions are replaced by the energy and species balances at a given transversal section of the enclosure. With this procedure and the fact that the quantities of heat and mass flowing through the horizontal section are equal to zero, the following expressions for the energy and species balances are obtained:

$$\int_{-1/2}^{+1/2} \left( \frac{\partial T}{\partial y} + D_u \frac{\partial S}{\partial y} \right) dx + \int_{-1/2}^{+1/2} \frac{\partial \Psi}{\partial x} T \, dx = 0 \tag{32}$$

$$\int_{-1/2}^{+1/2} \left( \frac{\partial S}{\partial y} + S_r \frac{\partial T}{\partial y} \right) dx + Le \int_{-1/2}^{+1/2} \frac{\partial \Psi}{\partial x} S \, dx = 0 \tag{33}$$

After substituting Equations (25), (30), and (31) into Equations (32) and (33) and performing the integration, the constant temperature and concentration gradients along the *y*-direction, $C_T$ and $C_S$ are respectively obtained as:

$$C_T = \frac{c_0 \, \Psi_{0n} \left( a_T - a_s Le D_u + b Le^2 \Psi_{0n}^2 \right)}{\left( 1 + b \Psi_{0n}^2 \right) \left( 1 + b Le^2 \Psi_{0n}^2 \right) - D_u S_r \left( 1 - b Le \Psi_{0n}^2 \right)^2} \tag{34}$$

$$C_s = \frac{c_0 \, \Psi_{0n} \left( a_s Le - a_T S_r + b Le^2 \Psi_{0n}^2 \right)}{\left( 1 + b \Psi_{0n}^2 \right) \left( 1 + b Le^2 \Psi_{0n}^2 \right) - D_u S_r \left( 1 - b Le \Psi_{0n}^2 \right)^2} \tag{35}$$

where:

$$b = \frac{1}{2(1 - D_u S_r)}, \quad c_0 = \frac{\eta_2}{\sqrt{\eta_1}}, \quad \eta_2 = \frac{2 \sin(\theta_2/2)}{\theta_2} \left( 1 + \frac{\theta_2^2}{\theta_1^2} \right) \tag{36}$$

Substituting expressions of $C_T$ and $C_S$ in the $\Omega$ definition, Equation (24), we obtain the following polynomial equation:

$$b^2 Le^2 \, \Psi_{0n}^4 + 2 \, d_1 b^2 \, \Psi_{0n}^2 - 4 b^2 d_2 R_T^0 \, \Psi_{0n} + 1 = 0 \tag{37}$$

where: $d_1 = 1 + Le^2 + 2 Le D_u S_r$ $d_2 = Le^2 + N + Le \left( D_u + N S_r \right)$ and $R_T^0$ being defined as: $R_T^0 = R_T / R^0$ with $R_0 = \Omega^2 / c_0$.

Equation (37) is solved numerically using the Newton–Raphson method. In this way, the value of $C_T$ and $C_S$, and the stream function, temperature, and concentration fields can be obtained for any combination set of the controlling parameters $R_T$, $L_e$, $Da_e$, $D_u$, and $S_r$.

From Equation (37), it is found that a non-zero solution exists only beyond a threshold. Subsequently, the normalized threshold Rayleigh number which characterizes the onset of convective motion is obtained by deriving Equation (37) with respect to $R_T^0$ and setting the derivative $\frac{d\Psi_{0n}}{dR_T^0}$. After some algebra, it is found that:

$$R_{TC}^{sub,0} = \frac{Le^2 \Psi_{0nC}^3 + \left( 1 + Le^2 + 2 Le D_u S_r \right) \Psi_{0nC}}{Le^2 + N + Le(D_u + N S_r)} \tag{38}$$

where $\Psi_{0nc}$ is the critical finite stream function value at the bifurcation point, which constitutes a saddle-node point characterizing the subcritical bifurcation. The expression of $\Psi_{0nc}$ is obtained as:

$$\Psi_{0nC} = \frac{1}{Le\sqrt{3b}} \left[ \sqrt{b^2(1 + Le^2 + 2 Le D_u S_r)^2 + 3 Le^2} - b \left( 1 + Le^2 + 2 Le D_u S_r \right) \right]^{\frac{1}{2}} \tag{39}$$

The threshold of the subcritical convection is computed from Equation (38), where the relationship between the normalized critical Rayleigh number, $R_{TC}^{sub,0}$, and the critical Rayleigh number, $R_{TC}^{sub}$, which characterizes the onset of convective motion, is given by:

$$R_{TC}^{sub} = R^0 \, R_{TC}^{sub,0} \tag{40}$$

According to the temperature and concentration profiles, Equations (30) and (31), the local Nusselt and Sherwood numbers expressions, Equation (16), are reduced to:

$$Nu = \frac{1}{1 - c_0 C_T \Psi_{0n}} \tag{41}$$

$$Sh = \frac{1}{1 - c_0 C_S Le \Psi_{0n}} \tag{42}$$

In the above equations, the parameters $b_0$, $c_0$, and $R^0$ depend on the Darcy number, $Da_e$, as depicted in Figure 3a–c. From Figure 3a, when the Darcy number is very small $\left(Da_e \leq 10^{-6}\right)$ the variation of the constant $b_0$ tends asymptotically toward $b_0 = 1$ and the normalized stream function can be rewritten as: $\Psi_{0n} = \Psi_0$ which corresponds to the pure Darcy situation. Upon increasing the value of the Darcy number, $b_0$ decreases and tends asymptotically toward a constant value: $b_0 = [\cosh^2(\omega/2) + \cos^2(\omega/2)]/[\cosh(\omega/2) - \cos(\omega/2)]^2 = 0.793$, which corresponds to the clear fluid case. A similar trend is observed for the evolution of the constant $c_0$, as illustrated in Figure 3b, where the limiting values $c_0 = 0.636$ and $c_0 = 0.588$ correspond to the Darcy and the clear fluid media, respectively. In addition, the effect of $Da_e$ on the constant $R_0$ is depicted in Figure 3c. As expected, the graph indicates that the constant $R^0$ decreases towards a constant value $R^0 = \pi^3/2 = 15.581$ when $Da_e \to 0$, and to $R^0/Da_e = \omega^5 \sqrt{1 + [\cos(\omega/2)/\cosh(\omega/2)]^2}/[4\sin(\omega/2)]$ when $Da_e \to \infty$.

*4.1. Case of Darcy Flow $(Da_e << 1)$*

The Darcy situation can be deduced from the Brinkman model when $Da_e \to 0$. For this condition, Equation (29) indicates that the parameter $\theta_1$ tends to $+\infty$, and Equation (28) leads to $\tan(\theta_2/2) \to -\infty$, which allows to define the parameter $\theta_2$ as: $\theta_2 \approx \pi(1+2n)$, and from Equation (29), $\Omega \approx \theta_2 \approx \pi(1+2n)$. Thus, in these conditions, for a monocellular flow ($n = 0$), where $\Omega \approx \theta_2 \approx \pi$, the expressions of the constants: $\eta_0$, $\eta_1$, and $\eta_2$ are reduced to: $\eta_0 \approx 0$, $\eta_1 \approx 1$, $\eta_2 \approx 2/\pi$. Using the above expressions, the definition of $b_0$ is reduced to $b_0 = 1$ and $\Psi_{0n} = \Psi_0$.

The Darcy solution of the present problem (i.e., $Da_e = 0$) is obtained by introducing the above results into the expressions (Equations (25), (30) and (31)) presented in Section 4, which can be reduced to the following equations:

$$\Psi(x) = \Psi_0 \cos(\Omega x) \tag{43}$$

$$T(x,y) = C_T y - \frac{\Psi_0}{\Omega} \left( \frac{C_T - C_s D_u Le}{1 - D_u S_r} \right) \sin(\Omega x) + a_T x \tag{44}$$

$$S(x,y) = C_S y - \frac{\Psi_0}{\Omega} \left( \frac{C_S Le - C_T S_r}{1 - D_u S_r} \right) \sin(\Omega x) + a_S x \tag{45}$$

In this situation $C_T$, $C_S$, $\Psi_0$, $Nu$, and $Sh$ are evaluated from Equations (34), (35), (37), (41) and (42) respectively. The critical Rayleigh number is then given by:

$$R_{TC}^{sub} = R^0 \frac{Le^2 \Psi_{0C}^3 + (1 + Le^2 + 2Le D_u S_r) \Psi_{0C}}{Le^2 + N + Le(D_u + NS_r)} \tag{46}$$

where $R^0 = \Omega^3/c_0 \approx \pi^3/2$.

### 4.2. Case of Fluid Flow $(Da_e \gg 1)$

When $Da_e \to \infty$, the current Brinkman model is reduced to a parallel flow solution in a vertical cavity filled with a clear fluid.

For this condition, Equation (29) indicates that $\theta_1 \approx \theta_2$, and $\Omega^2 = Da_e\theta_2^4$ or $\Omega^2 = Da_e\omega^4$, where $\omega = \sqrt[4]{Ra_T(LeC_S - C_S)/(1 - S_r)}$ with $Ra_T = R_T/Da_e$ being the thermal Rayleigh number for a clear fluid medium.

In this regard, Equation (28) reduces to:

$$\tan\left(\frac{\omega}{2}\right) + \tanh\left(\frac{\omega}{2}\right) = 0 \tag{47}$$

For $n = 0$, the flow is monocellular and a numerical solution of Equation (47) is obtained as $\omega_0 \approx 4.73$. For $n \gg 1$, an approximate asymptotic solution is obtained as: $\omega_n = \pi(4n+3)/2$, where each value of $n$ ($n = 1, 2, 3, 4 \dots$) corresponds to a different convective mode. The expressions of the constants $\eta_0$, $\eta_1$, and $\eta_2$ stated in Equations (26) and (36) are reduced to:

$$\eta_0 = \frac{\cos(\omega/2)}{\cosh(\omega/2)}, \quad \eta_1 = 1 + \eta_0^2, \quad \eta_2 = \frac{4\sin(\omega/2)}{\omega} \tag{48}$$

In this way, the expressions of $\Psi(x)$, $T(x,y)$, and $S(x,y)$ corresponding to the pure fluid medium are given by:

$$\Psi(x) = \frac{\Psi_0}{1 - \eta_0}[\cos(\omega x) - \eta_0\cosh(\omega x)] \tag{49}$$

$$T(x,y) = C_T y - \frac{\Psi_0}{\omega(1-\eta_0)}\left(\frac{C_T - C_s D_u Le}{1 - D_u S_r}\right)[\sin(\omega x) - \eta_0\sinh(\omega x)] + a_T x \tag{50}$$

$$S(x,y) = C_S y - \frac{\Psi_0}{\omega(1-\eta_0)}\left(\frac{C_S Le - C_T S_r}{1 - D_u S_r}\right)[\sin(\omega x) - \eta_0\sinh(\omega x)] + a_S x \tag{51}$$

For the clear fluid limit, $C_T$, $C_S$, $\Psi_0$, $N_u$, and $Sh$ expressions can be easily derived from Equations (34), (35), (37), (41), and (42), respectively.

For this situation, the critical Rayleigh number, $R_{TC}^{sub}$, is expressed by:

$$Ra_{TC}^{sub} = \sqrt{b_0}Ra^0\frac{b_0 Le^2\Psi_{0C}^3 + (1 + Le^2 + 2LeD_uS_r)\Psi_{0C}}{Le^2 + N + Le(D_u + NS_r)} \tag{52}$$

where: $Ra^0 = \omega^4/c_0$, and

$$\Psi_{0C} = \pm\frac{1}{Le\sqrt{3bb_0}}\left[\sqrt{b^2(1 + Le^2 + 2LeD_uS_r)^2 + 3Le^2} - b\left(1 + Le^2 + 2LeD_uS_r\right)\right]^{\frac{1}{2}} \tag{53}$$

In the past, the parallel flow concept used to predict the asymptotic analytical solution presented in this paper was validated by many authors. Most of these validations were performed for the case of double diffusive convection without Soret and Dufour effects [22,27] or with only the Soret effect [36]. Figure 4 illustrates the effect of the enclosure aspect ratio on the stream function value at the center of the enclosure and the local and average heat and mass transfer rates for $R_T = 50$, $Le = 5$, $\xi = 1$ and $D_u = S_r = 0.2$, within a Darcy porous medium. From the Figure, it is observed that the flow intensity and the local Nusselt number increase monotonically and sharply at the beginning as the aspect ratio increases beyond 1 and then tend asymptotically toward constant values as the aspect ratio becomes very large ($A \geq 6$). The same trend is observed for the averaged Sherwood number. However, for the local Sherwood number, first it increases drastically with the aspect ratio and then passing through a maximum it drops toward a constant value. The constant values are reached at $A = 6$ and they become independent of the aspect ratio of the enclosure; they compare well with the analytical solution. For this reason, and to be more

conservative, most of the numerical results presented in this paper are obtained for $A = 0$ to ensure the validity of the asymptotic analytical solution in the core region of the enclosure.

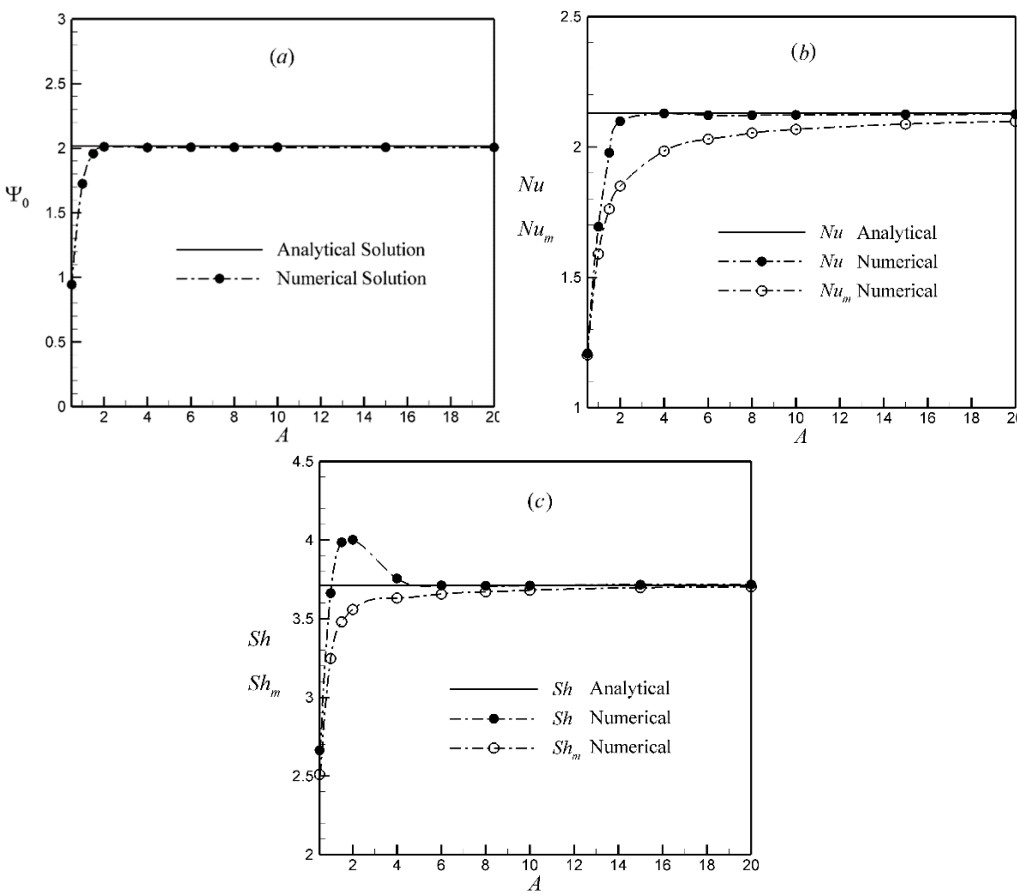

**Figure 4.** Enclosure aspect ratio effect on (**a**) the flow intensity, (**b**) and the heat, and (**c**) mass transfer rates for: $R_T = 50$, $Le = 5$, $\varepsilon = 1$ and $D_u = S_r = 0.2$.

## 5. Linear Stability Analysis

In this section, a two-dimensional stability analysis of the binary flow system is considered. The overall unsteady solution of the problem consists of a basic solution ($\Psi_b$, $T_b$, $S_b$) representing the pure diffusive rest state solution or the steady-state convective solution, and a perturbation solution ($\Psi_p$, $T_p$, $S_p$). The basic rest state solution is given by ($\Psi_b = 0$, $T_b = a_T x$, and $S_b$) $= a_s x$, and the basic convective steady state solution is obtained from the parallel flow asymptotic approach. For an infinite porous layer, this assumption allowed to define the perturbations, as follows:

$$\left.\begin{array}{l} \Psi_p(x,y,t) = \psi_0 e^{pt+iky} f(x) \\ T_p(x,y,t) = \theta_0 e^{pt+iky} g(x) \\ S_p(x,y,t) = \phi_0 e^{pt+iky} h(x) \end{array}\right\} \tag{54}$$

where $k$ is the wave number, $f(x)$, $g(x)$ and $h(x)$ are functions describing the perturbation profiles, and $p = p_r + ip_i$, is a complex number indicating the growth rate of the perturbation, $p_r$, and the oscillation frequency, $p_i$, and $\psi_0$, $\theta_0$, and $\phi_0$ are unknown infinitesimal amplitudes.

Substituting Equation (54) into the governing Equations (6)–(8) and after dropping second-order nonlinear terms, it yields the following linear stability equations:

$$
\left.
\begin{aligned}
\left(D^2 - k^2 - Da_e\left(D^4 - 2k^2 D^2 + k^4\right)\right)F + R_T D\left(G - a_T a_S^{-1} H\right) &= -\xi p\left(D^2 - k^2\right)F \\
\left(C_T D - ik(a_T + D\theta_b)\right)F + ikgD\Psi_b + \left(D^2 - k^2\right)(G + D_u H) &= pG \\
\left(C_S D - ik(a_S + D\varphi_b)\right)F + ikhD\Psi_b + Le^{-1}\left(D^2 - k^2\right)(S_r G + H) &= \varepsilon p H
\end{aligned}
\right\}
\tag{55}
$$

with $F = \psi_0 f$, $G = \theta_0 g$, $H = \phi_0 h$ and $D = d/dx$.

The new perturbation boundaries conditions are defined as follows:

$$
x = \pm\frac{1}{2}: \quad F = \frac{\partial F}{\partial x} = 0, \quad \frac{\partial G}{\partial x} = 0, \quad \frac{\partial H}{\partial x} = 0
\tag{56}
$$

The above linear system (55) subject to boundary conditions Equation (56) is solved numerically using a finite element method based on the cubic Hermite element. The numerical procedure is described in detail in Mamou [58] and Mamou et al. [59]. Since there is a slight difference between the current problem and that analyzed in [58,59], some details are omitted. The discretized linear equations are assembled into a global eigenvalue system as follows:

$$
\begin{bmatrix}
[K_\psi] & -R_T[B_\psi] & R_T \frac{a_T}{a_S}[B_\psi] \\
-[B_\theta] & [K_\theta] & D_u[L_\theta] \\
-[B_\phi] & \frac{S_r}{Le}[L_\phi] & \frac{1}{Le}[K_\phi]
\end{bmatrix}
\left\{
\begin{matrix} F \\ G \\ H \end{matrix}
\right\}
= p
\begin{bmatrix}
-\xi[M_\psi] & 0 & 0 \\
0 & -[M_\theta] & 0 \\
0 & 0 & -\varepsilon[M_\phi]
\end{bmatrix}
\left\{
\begin{matrix} F \\ G \\ H \end{matrix}
\right\}
\tag{57}
$$

where $F$, $G$, and $H$ are the unknown eigenvectors of dimension $m = 2N_{ex} + 1$, where $N_{ex}$ is the number of elements in $x$ direction, and $[K_\psi]$, $[K_\theta]$ $[K_\phi]$, $[B_\psi]$, $[B_\theta]$, $[B_\phi]$, $[L_\theta]$, $[L_\phi]$, $[M_\psi]$, $[M_\theta]$ and $[M_\phi]$ are square matrices of dimension $m \times m$, whose elementary matrices are defined as follows:

$$
\left.
\begin{aligned}
[K_\psi]^e &= \int_{\Delta x_e}\left[\frac{d\mathcal{H}_j}{dx}\frac{d\mathcal{H}_l}{dx} + k^2\mathcal{H}_j\mathcal{H}_l + Da_e\left(\frac{d^2\mathcal{H}_j}{dx^2}\frac{d^2\mathcal{H}_l}{dx^2} + 2k^2\frac{d\mathcal{H}_j}{dx}\frac{d\mathcal{H}_l}{dx} + k^4\mathcal{H}_j\mathcal{H}_l\right)\right]dx, \\[2mm]
[B_\psi]^e &= \int_{\Delta x_e}\frac{d\mathcal{H}_j}{dx}\mathcal{H}_l\,dx, \quad [M_\psi]^e = \int_{\Delta x_e}\left(\frac{d\mathcal{H}_j}{dx}\frac{d\mathcal{H}_l}{dx} + k^2\mathcal{H}_j\mathcal{H}_l\right)dx, \\[2mm]
[K_\theta]^e &= \int_{\Delta x_e}\left(\frac{d\mathcal{H}_j}{dx}\frac{d\mathcal{H}_l}{dx} + k^2\mathcal{H}_j\mathcal{H}_l - ik\frac{\partial\Psi_b}{\partial x}\mathcal{H}_j\mathcal{H}_l\right)dx, \\[2mm]
[B_\theta]^e &= \int_{\Delta x_e}\left[C_T\frac{\partial\mathcal{H}_j}{\partial x} - ik\left(a_T + \frac{d\theta_T}{dx}\right)\mathcal{H}_j\right]\mathcal{H}_l\,dx, \quad [M_\theta]^e = \int_{\Delta x_e}\mathcal{H}_j\,\mathcal{H}_l\,dx, \\[2mm]
[K_\phi]^e &= \int_{\Delta x_e}\left(\frac{d\mathcal{H}_j}{dx}\frac{d\mathcal{H}_l}{dx} + k^2\mathcal{H}_j\mathcal{H}_l - ikLe\frac{\partial\Psi_b}{\partial x}\mathcal{H}_j\mathcal{H}_l\right)dx, \\[2mm]
[B_\phi]^e &= \int_{\Delta x_e}\left[C_S\frac{d\mathcal{H}_j}{dx} - ik\left(a_S + \frac{d\theta_S}{dx}\right)\mathcal{H}_j\right]\mathcal{H}_l\,dx
\end{aligned}
\right\}
\tag{58}
$$

### 5.1. Stability of the Rest State

The stability of the motionless state: ($\Psi = 0$, $T = ax$ and $S = ax$) is now considered. The methodology for obtaining the thresholds of various types of convective modes is described hereafter. The eigenvalue problem, Equation (57), in its general form is valid for any governing parameters value.

To explicitly determine the thresholds of stationary and oscillatory convection, the Galerkin method is the most suitable provided that the eigenvectors are predetermined through the numerical analysis given in Section 5.

Now assuming that the eigenvectors ($F$, $G$, $H$) are obtained from Equation (57) so they can be used as the weighing functions, and substituting the rest state solution ($\Psi = 0$,

$T_b = a_T x$, $S_b = a_s x$) in to the general stability Equation (55) and performing the Galerkin integration, the following scalar linear equations are obtained:

$$p\xi\psi_0 M_\psi + \psi_0 K_\psi = R_T\left(\theta_0 - \phi_0 a_T a_S^{-1}\right)B \tag{59}$$

$$p\theta_0 M_\theta - \psi_0 a_T L_\theta = -(\theta_0 + \phi_0 D_u)K_\theta \tag{60}$$

$$p\varepsilon\phi_0 M_\phi - \psi_0 a_S L_\phi = -Le^{-1}(\phi_0 + \theta_0 S_r)K_\phi \tag{61}$$

where $M_\psi$, $M_\theta$, $M_\phi$ $K$, $K_\theta$, $K_\phi$, $L_\theta$, $L_\phi$ and $B$ are constants which can be computed from the following Galerkin integrals:

$$\left.\begin{array}{l}
K_\psi = \int_{-1/2}^{1/2}\left[\left(\frac{dF}{dx}\right)^2 + k^2 F^2 + Da_e\left(\left(\frac{d^2F}{dx^2}\right)^2 + 2k^2\left(\frac{dF}{dx}\right)^2 + k^4 F^2\right)\right]d\,x, \\[10pt]
K_\theta = \int_{-1/2}^{1/2}\left(\left(\frac{dG}{dx}\right)^2 + k^2 G^2\right)dx, \quad K_\phi = \int_{-1/2}^{1/2}\left(\left(\frac{dH}{dx}\right)^2 + k^2 H^2\right)dx, \\[10pt]
L_\theta = \int_{-1/2}^{1/2} ikGF dx, \quad L_\phi = \int_{-1/2}^{1/2} ikHF dx, B = \int_{-1/2}^{1/2}\frac{d\,G}{d\,x}F\,dx, \\[10pt]
M_\psi = \int_{-1/2}^{1/2}\left(\left(\frac{dF}{dx}\right)^2 + k^2 F^2\right)dx, M_\theta = \int_{-1/2}^{1/2} G^2 dx, M_\phi = \int_{-1/2}^{1/2} H^2 dx
\end{array}\right\} \tag{62}$$

with $K_\theta = K_\phi = K$, $L_\theta = L_\phi = L$ and $M_\theta = M_\phi = M$.

Substituting Equations (60) and (61) into Equation (59) we readily arrive to the following dispersion relationship:

$$\gamma\,\gamma_\psi\xi\varepsilon\,Le\left(\frac{p}{\gamma}\right)^3 + p_2\left(\frac{p}{\gamma}\right)^2 - p_1\left(\frac{p}{\gamma}\right) - p_0 = 0 \tag{63}$$

where

$$\left.\begin{array}{l}
p_0 = R_T^0\left(a_T a_S^{-1} - Le\right) - (1 - D_u S_r) \\[6pt]
p_1 = R_T^0 a_T Le(\varepsilon - 1) - (\varepsilon Le + 1) - (1 - D_u S_r)\gamma_\psi\gamma\xi \\[6pt]
p_2 = \varepsilon Le(1 + \gamma_\psi\gamma\xi) + \gamma_\psi\gamma\xi \\[6pt]
R_T^0 = \frac{R_T}{R^0}, R^0 = \frac{K_\psi K}{BL}, \gamma_\psi = \frac{M_\psi}{K_\psi}, \gamma = \frac{K}{M}
\end{array}\right\} \tag{64}$$

From Equation (63), the onset of overstabilities and stationary convection can be determined.

### 5.1.1. Onset of Stationary Convection

The determination of the thresholds of stationary and overstable convection are discussed. In general, the threshold of stationary convection is obtained when the marginal stability occurs ($p = 0$). After introducing the boundary conditions in the general linear system, Equation (57), the eigenvalue problem can be reduced to:

$$[E - \lambda I]\{F\} = 0 \tag{65}$$

with $E = [K_\psi]^{-1}[K]^{-1}[B_\psi][B]$ and $\lambda = \frac{1 - S_r}{R_T(a_T - a_S Le)}$, where $[I]$ is the identity matrix and $F$ is the eigenvector. The above equation has a nontrivial solution, ($\{F\} \neq 0$), only when the determinant of $[E - \lambda I]$, which yields $m$ eigenvalues which can be reorganized as $\lambda_1 \leq \lambda_2 \leq \ldots \leq \lambda_m$, and their corresponding eigenfunctions are given by $\{F\}_i$ where $i = 1, 2, \ldots, m$. Thus, from Equation (65), the threshold for stationary convection is given by:

$$R_{TC}^{sup} = R^0\frac{1 - S_r}{a_T - a_S Le} \tag{66}$$

where the constant $R^0$ takes positive values ($R^0 = 1/\lambda_m$) for $L_e < a_T a_S^{-1}$ and negative values ($R^0 = 1/\lambda_1$) for $L_e > a_T a_S^{-1}$. When $S_r = D_u = 0$, the above supercritical Rayleigh number expression is reduced to that reported by Mamou et al. [28]. For an infinite vertical Brinkman layer, the constant $R^0$ is a function of the effective Darcy number, $Da_e$, as also demonstrated by Mamou et al. [28] and Joly et al. [36]. The Darcy situation is obtained for $Da_e \leq 10^{-6}$ with slip boundary conditions where the constant $R^0$ is given by $R^0 = +105.33(-105.33)$ for $Le < a_T a_S^{-1} \left(Le > a_T a_S^{-1}\right)$, these values were reported by Mamou et al. [22] and Joly et al. [36]. The expression in Equation (66) can be obtained from Equation (63) when $p = 0$ (i.e., $p_0 = 0$).

### 5.1.2. Onset of Oscillatory Convection

The marginal state of overstability corresponds to the condition $p_r = 0$ (i.e., $p = ip_i$). Substituting the relation $p = ip_i$ in Equation (63), and after separating the imaginary and real parts, we find two expressions of $p_i$ as follows:

$$p_i^2 = -\gamma^2 \frac{p_0}{p_2}, \; p_i^2 = -\gamma \frac{p_1}{\varepsilon \, Le \, \gamma_\psi \xi} \tag{67}$$

The critical Rayleigh number $R_{TC}^{over}$, which characterizes the onset of oscillatory convection is obtained by equalization the two expressions stated in Equation (67), this leads to the critical Rayleigh number of the onset of the overstable regime as:

$$R_{TC}^{over} = R^0 \frac{(1-S_r)\varepsilon Le\gamma_\psi\gamma\xi - a_S\left[(\varepsilon Le+1)+(1-D_uS_r)\gamma_\psi\gamma\xi\right]\left[(\varepsilon Le+1)\gamma_\psi\gamma\xi+\varepsilon Le\right]}{(a_T-a_S Le)\,\varepsilon Le\gamma_\psi\gamma\xi - a_T a_S Le(\varepsilon-1)\left[(\varepsilon Le+1)\gamma_\psi\gamma\xi+\varepsilon Le\right]} \tag{68}$$

For the case where ($\xi = 0$), the critical Rayleigh number is reduced to:

$$R_{TC}^{over} = R^0 \frac{\varepsilon Le + 1}{a_T Le(\varepsilon - 1)} \tag{69}$$

The existence of the oscillatory convection mode is only possible when the condition $p_1^2 + p_0 < 0$ is satisfied, i.e., $R_{TC}^{over} < R_T < R_{TC}^{osc}$, where the value of $R_{TC}^{osc}$ is determined when $p_1^2 + p_0 = 0$. Additionally, and for the same situation ($\xi = 0$), the critical Rayleigh number, $R_{TC}^{osc}$, characterizing the upper limit of the oscillatory convection regime, where transition from oscillatory to stationary mode occurs, is obtained as follows:

$$R_{TC}^{osc} = R^0 \left( \frac{\varepsilon(a_S Le - a_T)}{2a_S a_T^2 Le(\varepsilon - 1)^2} + \frac{1 + \varepsilon Le}{a_T Le(\varepsilon - 1)} - \frac{\sqrt{\Delta}}{2[a_T Le(\varepsilon - 1)]^2} \right) \tag{70}$$

where: $\Delta = \varepsilon^2 Le^2 \left(a_T a_S^{-1} - Le\right)^2 - 4\,a_T\,\varepsilon\,Le^2(\varepsilon-1)\left[a_T a_S^{-1} - D_u Le + \varepsilon\,Le\left(S_r a_T a_S^{-1} - Le\right)\right]$.

The value of $R_{TC}^{over}$ and $R_{TC}^{osc}$ are also computed numerically from Equation (57) when $p_r = 0$ and $p_i \neq 0$, respectively, and minimized according to the optimal wavenumber.

### 5.2. Stability Analysis of the Convective State: Hopf Bifurcation

The stability of the basic finite amplitude convection solution, which is given by the parallel flow assumption developed in Section 4, is considered. Thus far, it is well known that when the Rayleigh number is increased above a critical value, the convective flow becomes oscillatory and the onset of oscillation is marked by a critical value known as the threshold of Hopf bifurcation. In order to find the threshold of a Hopf bifurcation, $R_{TC}^{Hopf}$, a stability analysis of the flow pattern is required. Table 3 shows the numerical results obtained using a non-uniform sinusoidal mesh. The effect of the grid size on the threshold of Hopf bifurcation is illustrated for an infinite vertical layer with $Da_e = 10^{-4}$, $Le = 2$, $S_r = D_u = 0.1$, $\xi = 0$, and $\varepsilon = 1$. The aim of the grid sensitivity study is to determine the best compromise between the accuracy of the results and the computational time. The results

are presented in terms of the critical Rayleigh number, $R_{TC}^{Hopf}$, critical wave number, $A_C$, and the oscillation frequency, $f_r$. The critical Rayleigh number value is seen to be accurate enough with a grid size of 32 finite elements. For $Da_e = 10^{-4}$, the Brinkman viscous layer is very thin and requires grid point clustering near walls.

**Table 3.** Effect of the grid size on the threshold of the Hopf bifurcation in an infinite vertical layer for $Da_e = 10^{-4}$, $Le = 2$, $S_r = D_u = 0.1$, $\xi = 0$, and $\varepsilon = 1$.

| Grid Size | 4 | 8 | 12 | 16 | 32 | 64 |
|---|---|---|---|---|---|---|
| $R_{TC}^{Hopf}$ | 1454.37 | 872.54 | 866.39 | 864.33 | 863.52 | 863.43 |
| $A_C$ | 2.73 | 2.90 | 2.91 | 2.91 | 2.89 | 2.91 |
| $f_r$ | 5.50 | 4.34 | 4.32 | 4.31 | 4.34 | 4.32 |

The effects of the Soret and Dufour parameters, $S_r$ and $D_u$, on the threshold of the Hopf bifurcation are shown in Table 4 for $Da_e = 10^{-4}$, $Le = 2$, $\xi = 0$, and $\varepsilon = 1$, with $S_r$ and $D_u$ varying from $-1$ to 0.6 and from 0 to 1, respectively. For this condition, an increase in $S_r$ delays the threshold of Hopf bifurcation up to $S_r = 0.5$ where $R_{TC}^{Hopf} \to \infty$. When $S_r > 0.5$, $R_{TC}^{Hopf}$ decreases with the increase in $S_r$, i.e., causing an early Hopf bifurcation. Thus, the Soret parameter has a stabilizing effect if $S_r < 0.5$, otherwise, it has a destabilizing effect. Furthermore, from the right side of Table 4, it is clear that the critical Rayleigh number $R_{TC}^{Hopf}$ decreases considerably with the increase in $D_u$. This follows from the fact that any increase in the Dufour parameter results in a destabilizing effect and enhances the convective flow. In general, the stability analysis leads to two conjugate solutions at the onset of the Hopf bifurcation. The perturbation flow patterns depicted in Figure 5 (for $Da_e = 10^{-4}$, $Le = 2$, $D_u = S_r = 0.1$, $\xi = 0$ and $\varepsilon = 1$) show that the two solutions (with negative and positive values of $p_i$) are mirror images of each other and once superposed could lead to traveling waves in the vertical direction.

**Table 4.** Effects of the Soret, $S_r$, and Dufour, $D_u$, parameters on the threshold of the Hopf bifurcation in an infinite vertical layer for $Da_e = 10^{-4}$, $Le = 2$, $S_r = D_u = 0.1$, $\xi = 0$, and $\varepsilon = 1$.

| | $D_u$ | | | | $S_r$ | | |
|---|---|---|---|---|---|---|---|
| $S_r$ | $R_{TC}^{Hopf}$ | $A_c$ | $f_r$ | $D_u$ | $R_{TC}^{Hopf}$ | $A_c$ | $f_r$ |
| $-1.0$ | 582.08 | 2.91 | 4.33 | | | | |
| $-0.6$ | 635.00 | 2.91 | 4.33 | | | | |
| $-0.2$ | 748.39 | 2.91 | 4.33 | 0.0 | 873.12 | 2.91 | 4.33 |
| 0.0 | 873.12 | 2.91 | 4.33 | 0.2 | 727.60 | 2.91 | 4.33 |
| 0.2 | 1164.16 | 2.91 | 4.33 | 0.6 | 545.70 | 2.91 | 4.33 |
| 0.6 | 1746.26 | 2.91 | 4.33 | 1.0 | 436.56 | 2.91 | 4.33 |

The effect of the acceleration parameter, $\xi$, on the onset of Hopf bifurcation is presented in Table 5 for $Da_e = 10^{-4}$, $Le = 2$, $D_u = S_r = 0.1$ and $\varepsilon = 1$. These results indicate clearly that when the acceleration parameter, $\xi$, increases from the 0 to 0.1, both the critical Rayleigh number, $R_{TC}^{Hopf}$, and wavelength, $A_C$, increase and the oscillation frequency, $f_r$, decreases. Thus, the acceleration parameter has a strong stabilizing effect on the onset of Hopf bifurcation, which delays the appearance of the oscillatory flows. These results are similar to those presented by Mamou [24] for double diffusive convection in the absence of both Soret and Dufour effects.

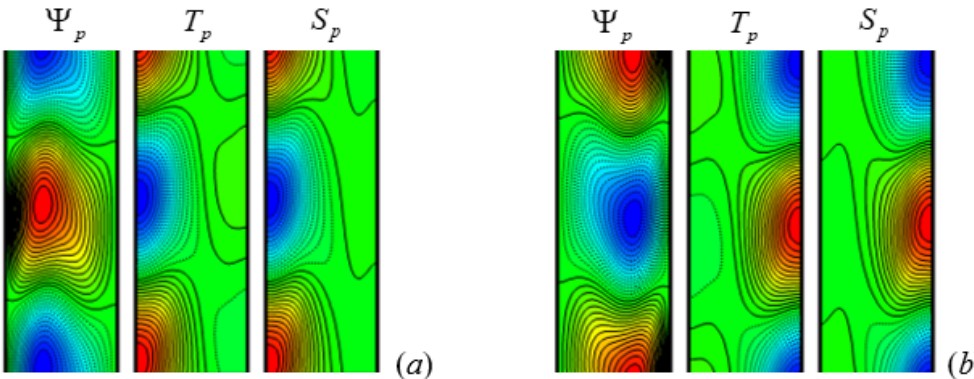

**Figure 5.** Perturbation profiles $\Psi p$, $T_p$, and $S_p$ at the threshold of Hopf bifurcation, $R_{TC}^{Hopf}$, for $Da_e = 10^{-4}$, $Le = 2$, $D_u = S_r = 0.1$, $\xi = 0$ and $\varepsilon = 1$: $R_{TC}^{Hopf} = 863.44$ and $A_C = 2.91$, (**a**) $p_i = 27.13$, and (**b**) $p_i = -27.13$.

**Table 5.** Effects of acceleration parameter $\xi$ on the threshold of the Hopf bifurcation in an infinite vertical layer for $Da_e = 10^{-4}$ $Le = 2$ $D_u = S_r = 0.1$, and $\varepsilon = 1$.

| $\xi$ | $R_{TC}^{Hopf}$ | $f_r$ | $A_c$ |
|---|---|---|---|
| 0 | 863.49 | 2.92 | 4.32 |
| $10^{-3}$ | 872.09 | 2.94 | 4.28 |
| $5 \times 10^{-3}$ | 913.70 | 3.06 | 4.16 |
| $10^{-2}$ | 982.20 | 3.27 | 3.97 |
| $5 \times 10^{-2}$ | 2114.62 | 7.99 | 2.08 |
| $10^{-1}$ | 3832.36 | 16.73 | 1.20 |

## 6. Results and Discussion

The main objective of the present investigation is to examine the influence of the Soret and Dufour effects on the convective flow and on heat and mass transfer rates near the onset of convection, and on the threshold of supercritical, overstable, and Hopf bifurcation convection. The Soret and Dufour parameters are varied within the ranges: $-1 < S_r < +1$ and $0 < D_u < +1$. As previously mentioned, this investigation is focused on the situation where the resultant of the thermal and solutal buoyancy forces is zero in the pure diffusive regime ($N = -a_T/a_S$).

The effect of the Soret and Dufour parameters on the stream function, vertical velocity, temperature, and concentration profiles at the mid-height of the porous layer (i.e., $y = 0$) are illustrated in Figures 6 and 7 for $0 < D_u < +1$, $Da_e = 1$, $Le = 10$, $R_T = 10^{-4}$, $\xi = 0$, and $\varepsilon = 1$. From these figures, it is clear that the asymptotic analytical solution, which is depicted in solid lines, is in good agreement with the numerical results presented by black circles. Figure 6a indicates that the flow intensity is clearly decreasing with the increase in the Soret effect and therefore reduces the flow circulation intensity inside the cavity, where the flow velocity for $S_r = 0.8$ is smaller than that corresponding to the case of double diffusive convection ($S_r = 0$), as shown in Figure 6b. At the same time, a little increase is observed in the flow intensity and the velocity magnitude with an increase in the Dufour effect, as shown in Figure 7a,b. Figure 6c shows that the effect of the Soret parameter causes a slight increase in the temperature difference between the two walls, which is clearly noticeable at $S_r = 0.8$. The effect of the Soret parameter on the concentration profiles, as illustrated in Figure 6d, causes a significant concentration difference decrease across the vertical walls as the Soret parameter is increased. The Dufour effect on the temperature profiles is depicted in Figure 7c, and its influence appears very similar to the effect of the Soret parameter on the concentration profiles. The presence of the Soret effect creates a solute deficit and surplus on the right and left walls, respectively, which becomes important with high values of $S_r$. Furthermore, the presence of the Dufour effect creates a heat deficit and surplus on

the right and left walls, respectively, which become enhanced as well at high values of $D_u$. However, for this particular situation, it appears that the Dufour parameter has no effect on the concentration profile, as depicted in Figure 7d.

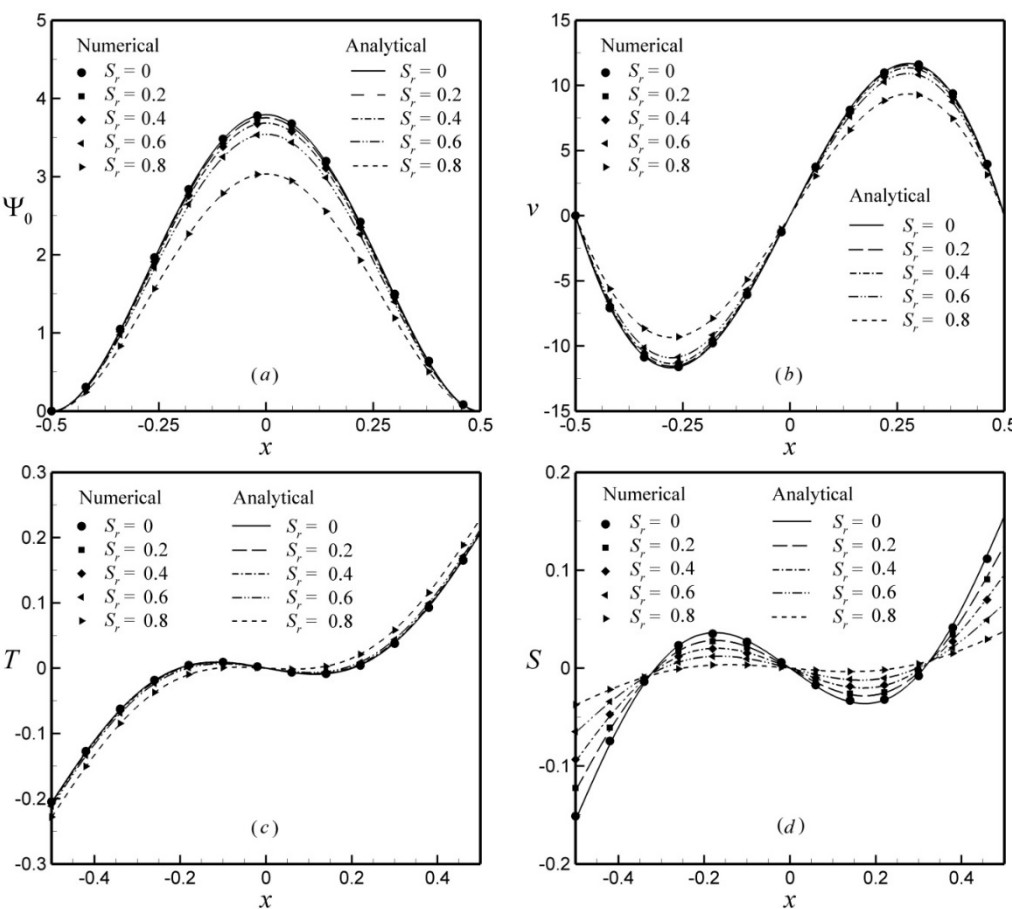

**Figure 6.** Stream function (**a**), velocity (**b**), temperature (**c**), and concentration (**d**) profiles in *x*-direction at mid-height of the enclosure for $A = 10$, $Da_e = 1$, $Le = 10$, $R_T = 10^4$, $\xi = 0$, $\varepsilon = 1$, $D_u = 0$ and various values of $S_r$.

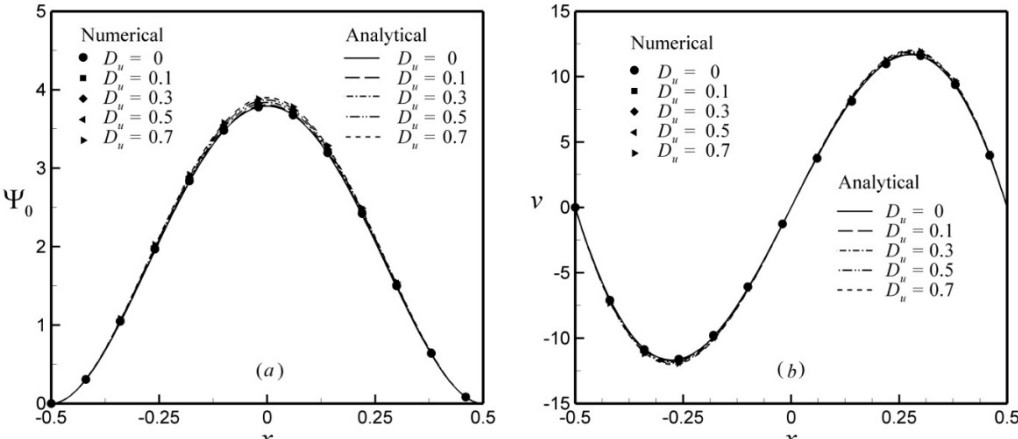

**Figure 7.** *Cont.*

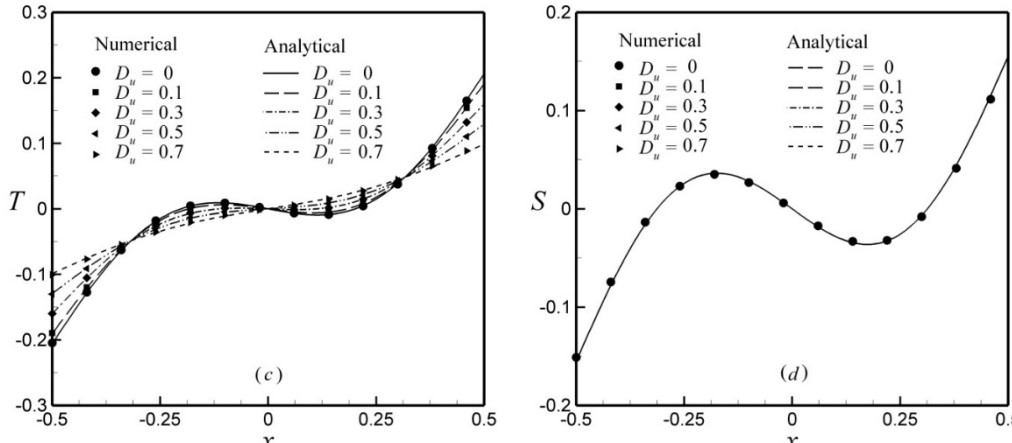

**Figure 7.** Stream function (**a**), velocity (**b**), temperature (**c**), and concentration (**d**) profiles in $x$-direction at mid-height of the enclosure for $A = 10$, $Da_e = 1$, $Le = 10$, $R_T = 10^4$, $\xi = 0$, $\varepsilon = 1$, $S_r = 0$ and various values of $D_u$.

Figures 8 and 9 show the effects of the Rayleigh number and the Soret and Dufour parameters on the stream function at the center of the cavity, $\Psi = 0$, and on the heat and mass transfer rates, $Nu$ and $Sh$, respectively, for $Le = 2$, $Da_e = 10^{-4}$, $\xi = 0$ and $\varepsilon = 1$. The curves depicted in the graphs are the predictions of the nonlinear parallel flow theory, which is developed in Section 4. Figures 8a and 9a show that for a given value of $S_r$ and $D_u$, there exists a Rayleigh number, $R_{TC}^{sub}$, for the onset of subcritical finite amplitude convection, where its value increases with the increase in the Soret parameter, while it decreases when the Dufour number increases. The strength of the convection intensity, $\Psi_0$, increases monotonously with the Rayleigh number, $R_T$, increase, and it becomes enhanced when the Soret parameter takes negative values, while it weakens for positive values. Furthermore, the strength of convection increases with an increase in the Dufour number. According to the results depicted in Figures 8 and 9b,c, it is obvious that when the Rayleigh number tends toward large values, whatever the values of the Soret and Dufour parameters are, both the heat and mass transfer rates tend asymptotically toward a constant value ($Nu = Sh = 4.88$), which depends on the Darcy number. From a physical point of view, this limit cannot be sustained as the flow becomes unstable for high values of $R_T$ and transition toward turbulence is imminent. For negative values of $S_r$, $Sh$ seems to exceed the asymptotic limit for small values of $R_T$, where it increases with the increase in $R_T$ until it reaches a maximum value, $Sh = 13.87$ at $R_T = 36$ for $S_r = -0.6$, and then drops later on towards the asymptotic value for large $R_T$. For $S_r = -0.4$, the maximum value of $Sh$ decreases to $Sh = 6.61$, and is reached at a slightly high Rayleigh number, $R_T = 50$.

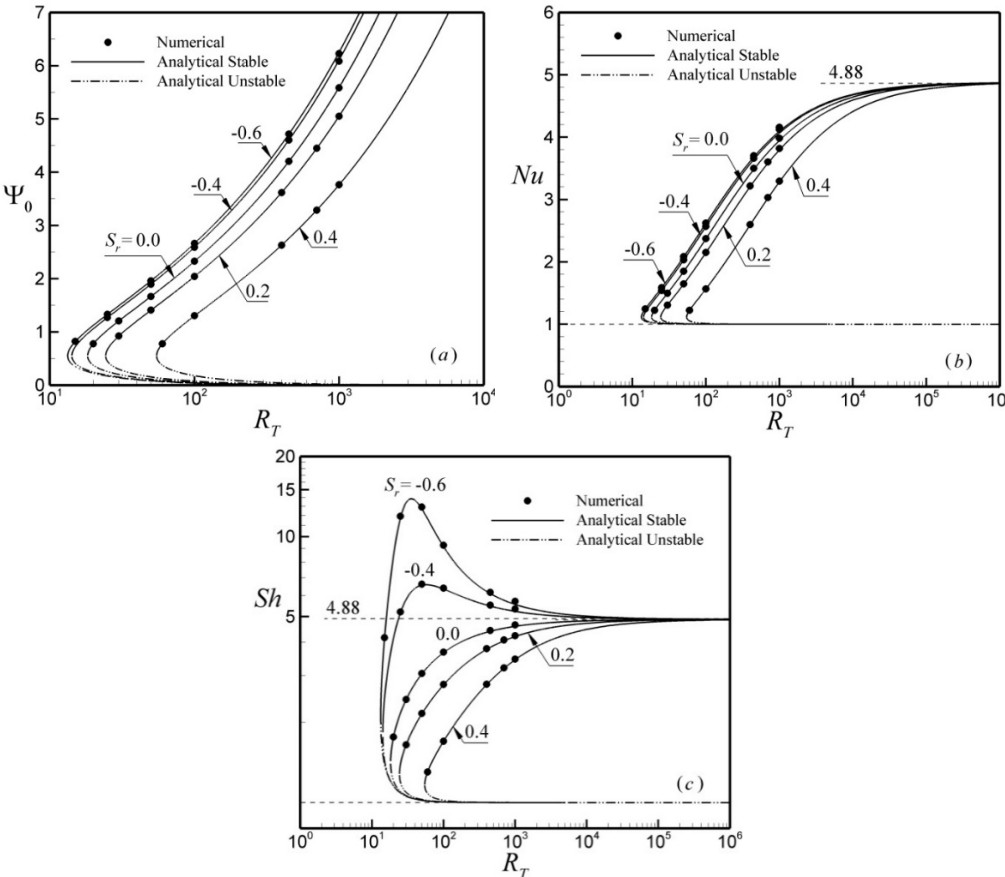

**Figure 8.** Bifurcation diagram as a function of $R_T$ and $S_r$ for $Da_e = 10^{-4}$, $Le = 2$, $\xi = 0$, and $\varepsilon = 1$: (**a**) flow intensity, $\Psi_0$, (**b**) local Nusselt number, $Nu$, and (**c**) local Sherwood number, $Sh$.

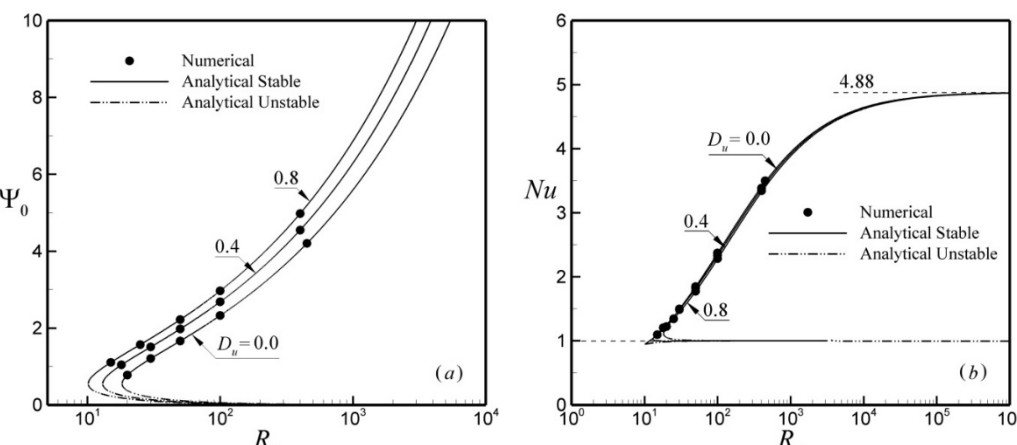

**Figure 9.** *Cont.*

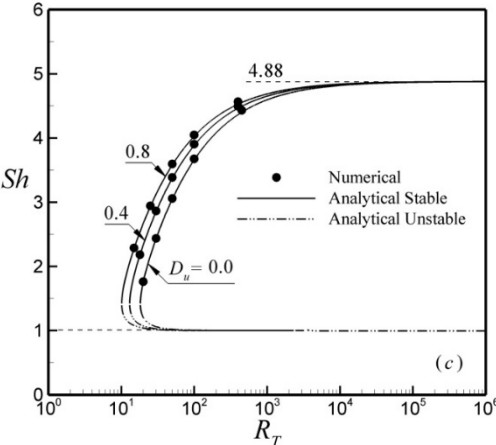

**Figure 9.** Bifurcation diagram as a function of $R_T$ and $D_u$ for $Da_e = 10^{-4}$, $Le = 2$, $\xi = 0$, and $\varepsilon = 1$: (**a**) flow intensity, $\Psi_0$, (**b**) local Nusselt number, $Nu$, and (**c**) local Sherwood number, $Sh$.

The effects of the Soret and Dufour parameters on the evolution of the subcritical Rayleigh number with the Lewis number are illustrated in Figure 10 for the case of a monocellular flow. These results were evaluated from the analytical nonlinear theory model by searching numerically for the value of $R_T^0$ for which the inverse of the derivative of $\Psi_{0n}$ with respect to $R_T^0$ is equal to zero. Figure 10a,b indicate that, for a given value of the Soret and Dufour parameters, $R_{TC}^{sub,0} \to \infty$ when $Le \to Le^*$, at which the convection is absent. From Equation (38), the subcritical Rayleigh number tends to infinity for $Le \to Le^*$, which is defined as follows:

$$Le^* = \frac{-(D_u + NS_r) + \sqrt{(D_u + NS_r)^2 - 4N}}{2} \tag{71}$$

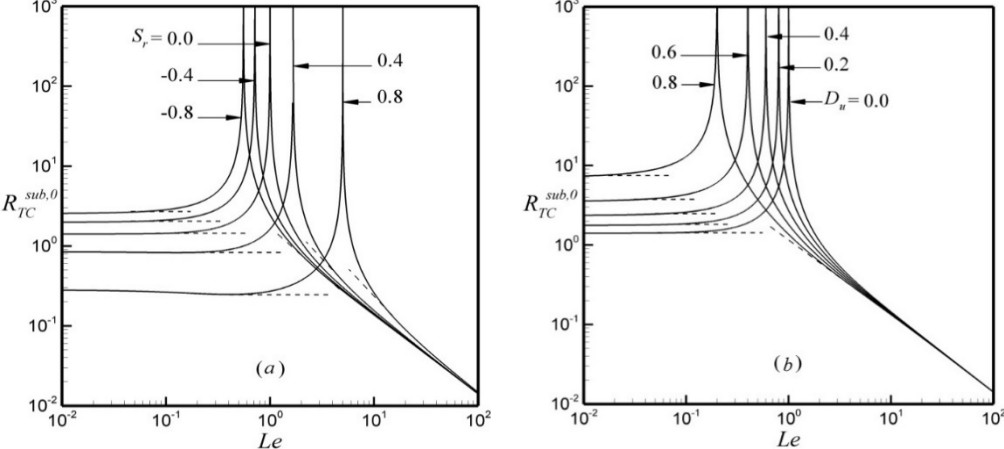

**Figure 10.** Subcritical Rayleigh number $R_{TC}^{sub,0}$ as a function of the Lewis number, $Le$, and various values of (**a**) $S_r$ ($D_u = 0$) and (**b**) $D_u$ ($S_r = 0$).

According to the above expression, the value of the Lewis number $Le^*$ for which the system is stable depends on the values of the Soret and Dufour parameters; where it is equal to unity ($Le^* = 1$) for $S_r = D_u = 0$, and it increases and becomes superior to unity ($Le^* > 1$) when the Soret parameter takes positive values, while it decreases below unity for negative values and with the increase in the Dufour number. For a given value of $S_r$ and $D_u$, upon decreasing the value of the Lewis number below $Le^*$, the subcritical Rayleigh number decreases towards a constant value, while it tends to zero $R_{TC}^{sub,0} \to 0$ when the Lewis

number increases above $Le^*$. Figure 10a shows that the negative (positive) value of the Soret parameter has a stabilizing (destabilizing) effect. The value of $R_{TC}^{sub,0}$ increases (decreases) with a decrease (increasing) in the Soret parameter when $Le < Le^*$. However, when $Le > Le^*$, it has a destabilizing (stabilizing) effect and the value of $R_{TC}^{sub,0}$ decreases (increases) with the decrease (increase) in $S_r$. From Figure 10b, it is clear that when $Le < Le^*$, the subcritical Rayleigh number increases with the $D_u$ increase, the Dufour parameter increase thus has a stabilizing effect. When when $Le > Le^*$, $R_{TC}^{sub,0}$ decreases with the increasing of the Dufour parameter, the $D_u$ increase thus has a destabilizing effect. For large values of $Le$, the Soret and Dufour parameters have a weak effect on the $R_{TC}^{sub,0}$.

The influence of the Soret effect on the flow intensity and the heat and mass transfer rates is illustrated in Figure 11 for $Da_e = 10^{-4}$, $R_T = 10^3$, $Le = 2$, $\xi = 0$, $\varepsilon = 1$, and $D_u = 0$. Results are obtained for values of $S_r$ ranging from $-1$ to $1$, exclusively. As can be observed, a good agreement is obtained between the parallel flow approximation presented by solid lines and the numerical solution displayed by solid symbols. Very distant from the onset of convection, a large value of $S_r$ induces oscillatory flows and transition towards chaos. The absence of the convective solution is noticeable for $0.495 \leq S_r \leq 0.505$, where the rest state prevails since, for this range of $S_r$, $R_T = 10^3$ is below the Rayleigh number, $R_{TC}^{sub}$, Equation (40). At $S_r = 0.495$ and $0.505$, a bifurcation from conductive to convective regime occurs ($\Psi_0 = 0.755$, $Nu = 1.214$ and $Sh = 1.218$ for $S_r = 0.495$ and $\Psi_0 = -0.783$, $Nu = 1.229$ and $Sh = 1.225$ for $S_r = 0.505$). Figure 11a indicates that, as the Soret parameter increases above $0.5$, the strength of the convective motion increases progressively and it becomes more and more important as the value of the Soret parameter approaches unity. The local Nusselt and Sherwood numbers (Figure 11b,c) are increasing sharply from the bifurcation point with the increase in $S_r$, where the heat transfer rate is more important than the mass transfer rate ($Nu > Sh$). As the value of the Soret parameter is made smaller, the convective flow intensity and the heat transfer rate become relatively independent of the Soret parameter, while the heat transfer rate is increased with a decrease in the value of the Soret parameter. From Figure 11b,c, it can be clearly observed that the faster diffusive component is the solute for $S_r > 0.5$, where the heat transfer is due mainly by convection ($Nu > Sh$), while for $S_r < 0.5$, the solute transfer is dominated by convection ($Sh < Nu$) as the heat is now the faster diffusing component. In addition, Figure 11a shows that the convective flow changes its rotation direction from counter-clockwise to clockwise when the Soret parameter varies from $-1$ to $+1$; counterclockwise for $S_r < 0.5$ and clockwise for $S_r > 0.5$, as shown in the flow patterns in Figure 11a. Furthermore, Figure 12 displays the influence of the Dufour effect on the flow behavior, $\Psi_0$, and the heat and mass transfer rates, $Nu$ and $Sh$, respectively, for $Da_e = 10^{-4}$, $R_T = 10^3$, $Le = 2$, $\xi = 0$, $\varepsilon = 1$ and $S_r = 0$. A good agreement is noticed between the asymptotic and numerical solutions. Figure 12a shows that, the convective flow intensity increases as Dufour effect increases. At the same time, the increase in the Dufour effect caused a little increase and decrease on the local Sherwood and Nusselt numbers, respectively, see Figure 12b. When $D_u$ increases from $D_u = 0$ to $D_u = 1$, the solute transfer rate increases slightly, while the heat transfer rate dwindles progressively.

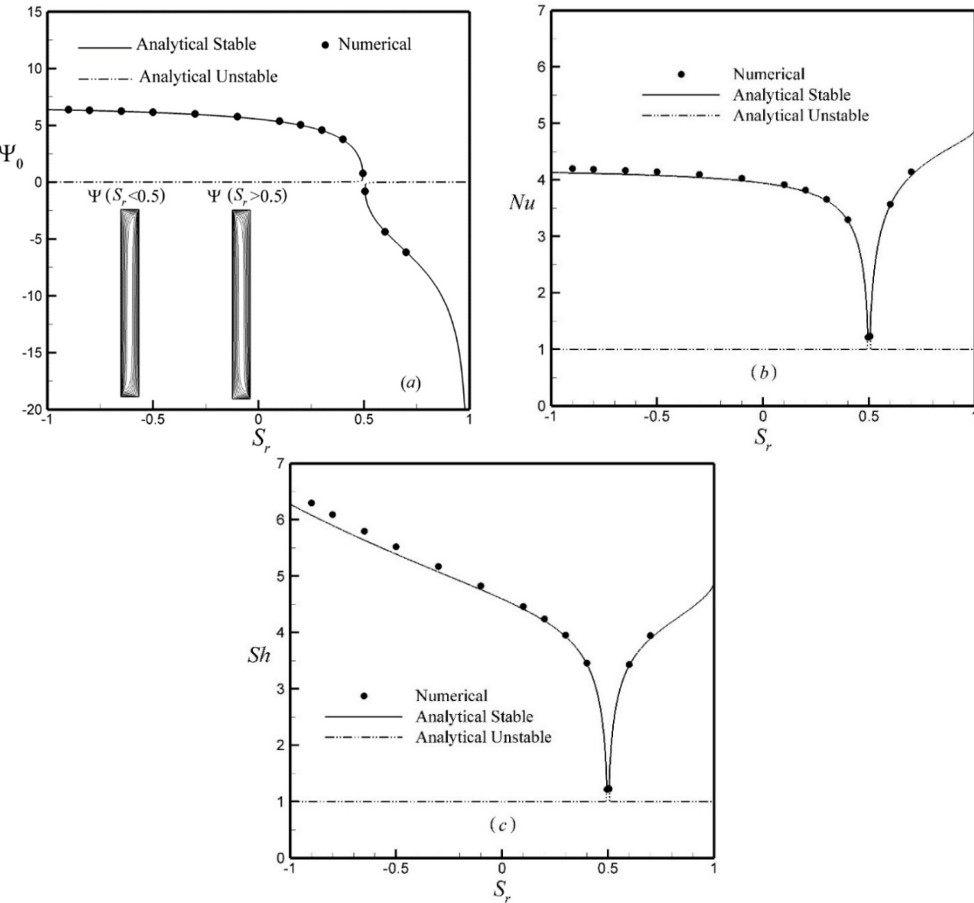

**Figure 11.** Effect of Soret parameter, $S_r$, on (**a**) stream function at the center of the cavity, $\Psi_0$, (**b**) local Nusselt number, $Nu$, and (**c**) local Sherwood number, $Sh$, for $Da_e = 10^{-4}$, $R_T = 10^3$, $Le = 2$, $\xi = 0$, $\varepsilon = 1$ and $D_u = 0$.

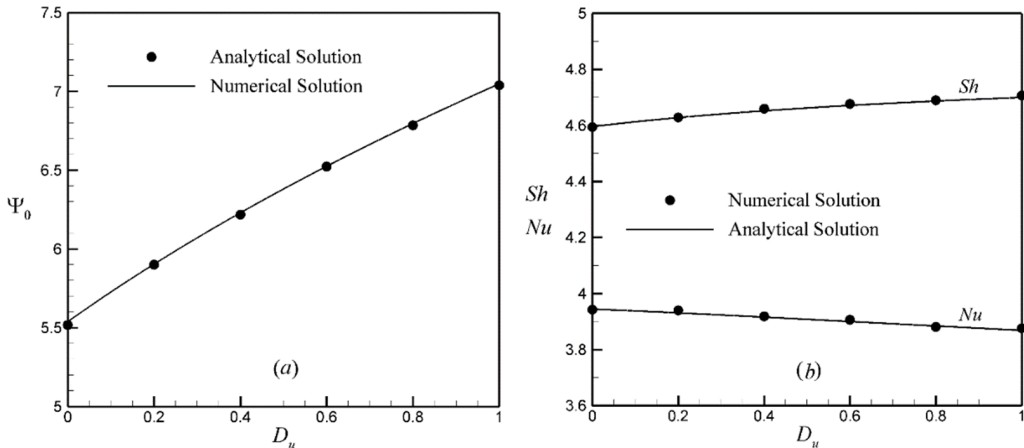

**Figure 12.** Effect of Dufour parameter, $D_u$, on (**a**) stream function at the center of the cavity, $\Psi_0$, (**b**) local Nusselt number, $Nu$, and local Sherwood number, $Sh$, for $Da_e = 10^{-4}$, $R_T = 10^3$, $Le = 2$, $\xi = 0$, $\varepsilon = 1$ and $S_r = 0$.

Figure 13a,b exemplify the effects of the Soret and Dufour parameters on the thresholds of bifurcations $R_{TC}^{sub}$, $R_{TC}^{sup}$ and $R_{TC}^{Hopf}$ for $Da_e = 10^{-4}$, $Le = 2$, $\xi = 0$ and $\varepsilon = 1$. The critical Rayleigh numbers for the onset of stationary convection, $R_{TC}^{sup}$, and for the onset of Hopf bifurcation, $R_{TC}^{Hopf}$, are predicted by the linear stability theory. However, the onset of

subcritical finite amplitude convection occurring at a Rayleigh number, $R_{TC}^{sub}$, is predicted by the present nonlinear model, Equation (40), such that upon decreasing the value of the Rayleigh number below $R_{TC}^{sub}$, the convective flow is at rest, which corresponds to the stable diffusive regime in which all perturbations decay (region I). In region (II), the linear theory predicts a stable rest state with possible finite amplitude convection according to the non-linear theory. In region (III) the system is unstable; any arbitrary dynamic perturbation can initiate a convective flow. Region (V) corresponds to the oscillatory finite amplitude convection that occurs right above the threshold for Hopf bifurcation. In general, it is seen from Figure 13a that the thresholds $\left( R_{TC}^{sub}, R_{TC}^{sup}, R_{TC}^{Hopf} \right)$ decrease sharply towards a constant value as the Soret parameter is decreased below $S_r = 0.5$. On the other hand, upon increasing the value of the Soret parameter above 0.5, the thresholds of bifurcation decrease monotonously toward $\left( R_{TC}^{sub}, R_{TC}^{sup}, R_{TC}^{Hopf} \right) \to 0$. It can be clearly observed that $\left( R_{TC}^{sub}, R_{TC}^{sup}, R_{TC}^{Hopf} \right) \to \infty$ when $S_r = 0.5$, which results in a stable system. This value varies dependently on the value of the Lewis number and the Dufour parameter which is expressed according to Equation (38) as follows:

$$S_r = 1 + \frac{D_u - 1}{Le} \tag{72}$$

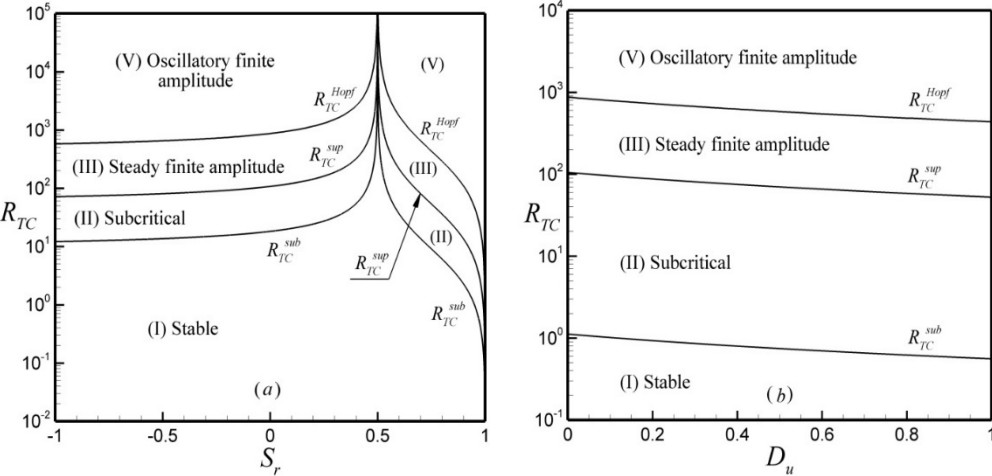

**Figure 13.** Critical Rayleigh numbers $R_{TC}^{sub}$, $R_{TC}^{sup}$, $R_{TC}^{Hopf}$ as functions of (**a**) Soret parameter $S_r(D_u = 0)$ and (**b**) Dufour parameter $D_u(S_r = 0)$, for $Da_e = 10^{-4}$, $Le = 2$, $\xi = 0$, and $\varepsilon = 1$.

For the case with $D_u = 0$ as considered in Figure 13a, the above expression reduces to $S_r = 1 - 1/Le$, i.e., the value of $S_r$ where $\left( R_{TC}^{sub}, R_{TC}^{sup}, R_{TC}^{Hopf} \right) \to \infty$ depends only on the value of the Lewis number. For Dufour induced convection, Figure 13b shows that the thresholds $\left( R_{TC}^{sub}, R_{TC}^{sup}, R_{TC}^{Hopf} \right)$ decrease monotonously upon an increase in the value of $D_u$. It follows that the steady parallel flow is destabilized earlier with an increase in $S_r$ and $D_u$. In other words, a decrease in the Soret and Dufour numbers results in a weakening of the convective flow and thus stabilizes the system.

Figure 14 shows the stability diagram in terms of $R_{TC}$ versus $Le$, as predicted by the linear stability analysis and the parallel flow model, Equations (40), (66) and (70), for $Da_e = 10^{-4}$, $D_u = S_r = 0.1$, $\xi = 0$ and $\varepsilon = 0.5$. In fact, the evolution of the curves depicted in the graph with the critical Rayleigh numbers, $R_{TC}^{sub}$, $R_{TC}^{sup}$ and $R_{TC}^{Hopf}$ is quite similar to that discussed in Figure 13a while studying the effect of Soret, $S_r$. As expected, the graph indicates that the critical Rayleigh numbers for the onset of subcritical, supercritical, and Hopf bifurcation convection decreases sharply toward a constant value as the Lewis number is decreased below unity. On the other hand, upon increasing the value of the

Lewis number above unity, the onset of convection decreases monotonously toward zero: $\left( R_{TC}^{sub}, R_{TC}^{sup}, R_{TC}^{Hopf} \right) \to 0$. In general, it is observed that $\left( R_{TC}^{sub}, R_{TC}^{sup}, R_{TC}^{Hopf} \right) \to \infty$ when $Le \to 1$, which results from the inexistence of any convection state. This result has been also reported by Mamou et al. [22], Amahmid et al. [27] and Rebhi et al. [39] while investigating the thresholds of bifurcation in a fluid and porous layer under various thermal and solutal boundary conditions with $\varepsilon = 1$. For a given value of the normalized porosity $\varepsilon = 0.5$, the linear stability theory predicts the existence of oscillating flow patterns within the overstable regime (zone (IV)), which is marked by the hatched area (i.e., delineated by $R_{TC}^{over}$ and $R_{TC}^{osc}$, Equations (69) and (70)). In the overstable region, convection is triggered from the rest state and it is amplified in an oscillatory manner.

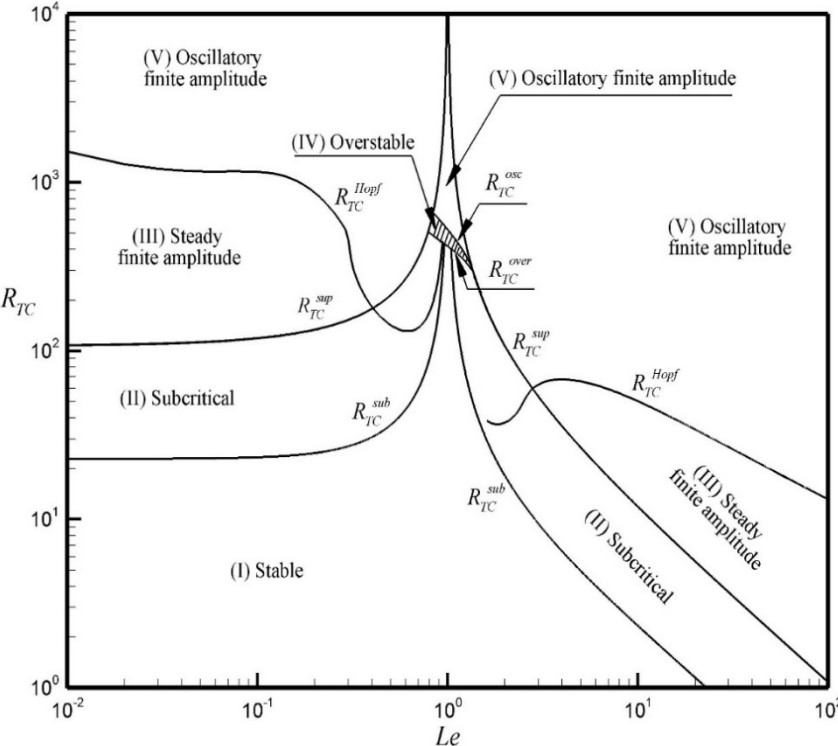

**Figure 14.** Stability diagram as a function of the Lewis number, *Le*, for $Da_e = 10^{-4}$, $D_u = S_r = 0.1$, $\xi = 0$, and $\varepsilon = 0.5$.

Figure 15 exemplifies the effect of the normalized porosity, $\varepsilon$, on the onset of Hopf bifurcation, $R_{TC}^{Hopf}$. The bifurcation diagram is presented in terms of $\Psi_0$, $Nu$ and $Sh$ as a function of $R_{TC}$ for $Da_e = 10^{-4}$, $Le = 2$, $\xi = 0$ and $D_u = S_r = 0.1$. In the graph, $\Psi_0$ is the flow intensity at the center of the cavity, $\Psi_0^\tau$ is the averaged flow intensity over a time period of the oscillation, $Nu$ and $Sh$ are the local Nusselt and Sherwood numbers at the mid-height of the cavity, and $Nu^\tau$ and $Sh^\tau$ are the time-averaged local Nusselt and Sherwood numbers, respectively. The computation of the time period averaged results is performed as follows: $f^\tau = \tau^{-1} \int_t^{t+\tau} f \, dt$, where $f$ stands for $\Psi_0$, $Nu$ and $Sh$, and $\tau$ is the oscillation time period. The curves depicted in the graphs are the predictions of the present analytical and numerical nonlinear theories. The solid lines correspond to the stable branches and the dot-dot-dashed lines to the unstable ones. The steady and unsteady numerical solutions of the full governing equations, obtained for $A = 10$, are shown by symbol (solid symbols for steady state solution and empty symbols with dashed lines for unsteady solution). In the steady state regime, a good agreement is observed between these two nonlinear theory results. The intersection between the solid symbols and the empty symbols curves represents the critical Rayleigh number for the onset of Hopf bifurcation, $R_{TC}^{Hopf}$, at which the transition from stationary to oscillatory convection occurs. In the graphs, only four data points are presented

for the oscillatory solutions. It is noticed that a further increase in the Rayleigh number beyond $R_{TC}^{Hopf}$ leads very quickly to periodic followed by chaotic oscillatory convective flows. At $\varepsilon = 0.6$, 0.8, and 1.0, it is found that $R_{TC}^{Hopf} = 75$, 305 and 910, respectively. These values were obtained by trial and error method by narrowing the gap where the transition occurs. For this value, the flow remains unicellular but the parallel nature of the streamlines is slightly distorted, which indicates the existence of a trail of small vortices in the core of the layer traveling along the vertical walls, as depicted in Figure 15c, while the time-averaged streamlines, $\Psi_m$, isotherms, $T_m$, and isoconcentrations, $S_m$, are displayed in Figure 15d for $R_T = 910$ and $\varepsilon = 1$. The critical Rayleigh number, $R_{TC}^{Hopf}$, predicted by the linear stability theory is obtained as 74.96, 314.27 and 863.44 for $\varepsilon = 0.6$, 0.8 and 1, respectively. A reasonable agreement between the different methods of solution is observed. This result indicates that the critical Rayleigh number, $R_{TC}^{Hopf}$, is strongly dependent on the value of the normalized porosity, $\varepsilon$, which plays the role of a stabilizing factor. A similar trend has been reported recently by Mamou et al. [22] while investigating the effect of the normalized porosity on double-diffusive convection induced by thermal and solutal gradients in a vertical porous layer.

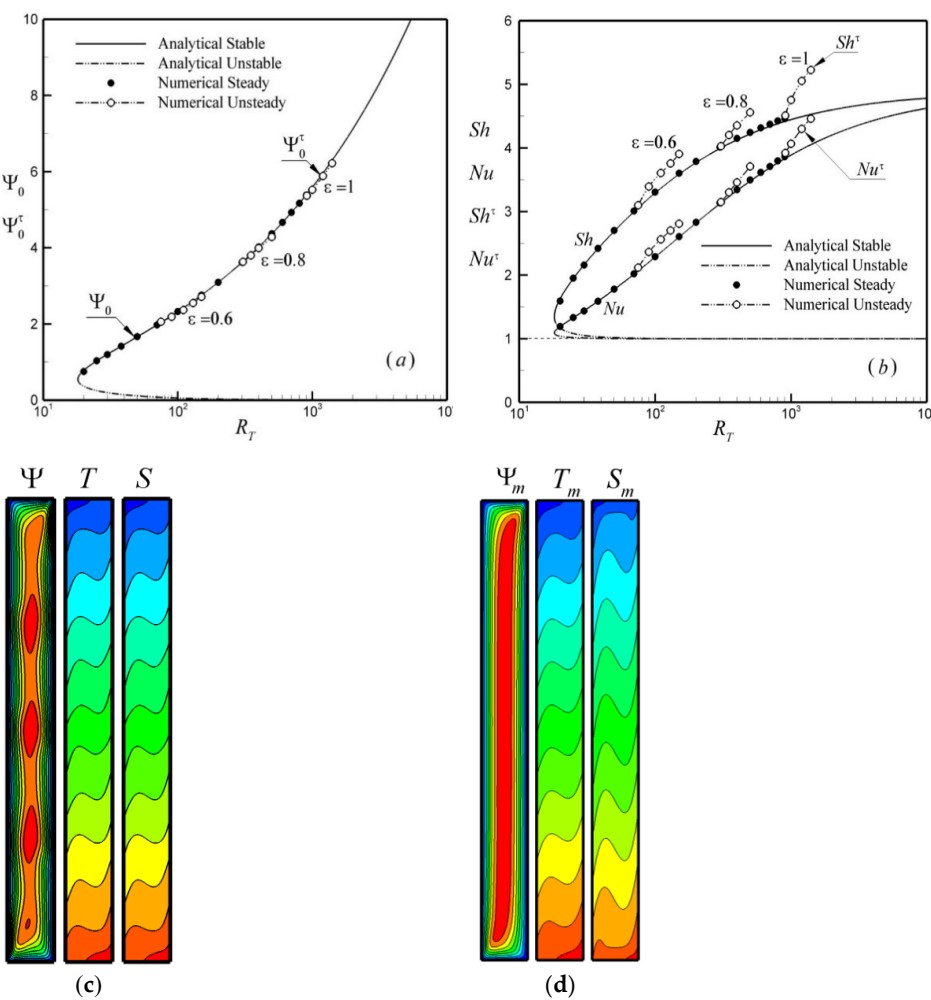

**Figure 15.** Bifurcation diagram in terms of (**a**) flow intensity, $\Psi_0$, and (**b**) local Nusselt, *Nu*, and Sherwood numbers, *Sh*, versus Rayleigh number, $R_T$, for $A = 10$, $Da_e = 10^{-4}$, $Le = 2$, $\xi = 0$ and $D_u = S_r = 0.1$, (**c**) Contours of stream (left), temperature (middle), and concentration (right) at a given instant of time, and (**d**) streamlines, isotherms and isoconcentrations of the time-period averaged solution for $R_T = 910$, $\varepsilon = 1$ and $\tau = 1.2$.

For a physical explanation of the transitional convective flow, unsteady solutions are presented in Figure 16a–d. The time histories of the flow intensity, $\Psi_0$, obtained for $R_T = 910$ and $\varepsilon = 1$, just above the critical Rayleigh number of Hopf bifurcation predicted by the linear stability theory, $R_{TC}^{Hopf} = 863.44$, is displayed in Figure 16a for a few cycles. The resulting oscillatory flow is found to be simply periodic. In Figure 16b–d, time-snapshots of the perturbations of the stream function, $\Psi_p$, temperature, $T_p$, and concentration, $S_p$, contours are displayed at different time steps during a time-period of oscillation, points (1)–(6). At each given time step, the stream function, temperature and concentration perturbation profiles are computed as $f_p = f^\tau - \tau^{-1} \int_t^{t+\tau} f dt$ where $f_p$ stands for $\Psi$, $T$ and $S$. The critical wave length and oscillatory frequency, which could be translated into a traveling wave speed, is obtained as $A_C = 2.5$ and $f_r = 5.63$ from the full numerical solution at $R_T = 910$, and as $A_C = 2.91$ and $f_r = 4.31$ from the linear stability analysis at $R_{TC}^{Hopf} = 863.44$. The small difference seen between the two approaches can be attributed to the finite aspect ratio considered, here $A = 10$. The wave length from the numerical solution is computed as the distance between two consecutive cells rotating in the same direction in the central part of the enclosure. As known, because of the relatively large value of $A_C = 2.91$ predicted by the linear stability analysis, an aspect ratio of $A_C = 10$ may be relatively small to simulate the infinite layer. It is expected that increasing the enclosure aspect ratio may improve the results, and the wavelength value variation may experience a jump as the convective perturbation cells increase oddly in numbers and the jump dwindles as we approach the situation of an infinite layer. In Figure 16b–d, the convection perturbation patterns are exemplified by two layers consisting of a series of small counter-rotating vortices traveling along the vertical walls in both ways, upwards near the right wall and downwards near the left wall. The vortices trail is seen to travel in a clockwise circulation. The vortices are seen to become weak as they negotiate their way through the end walls of the enclosure and regain their strength progressively later on as they quit the end regions. In general, it is seen from Figure 16b,d that the shape of the formed vortices is approximately the same as those predicted by the stability analysis when superposing the two conjugate solutions, Figure 5a,b.

A more complete view of the effect of the normalized porosity, $\varepsilon$, on the thresholds of bifurcation $\left( R_{TC}^{sub}, R_{TC}^{over}, R_{TC}^{osc}, R_{TC}^{sup}, R_{TC}^{Hopf} \right)$ is presented in Figure 17 for $Da_e = 10^{-4}$, $Le = 2$, and $D_u = S_r = 0.1$. The stability diagram is built according to the linear stability analysis and the parallel flow approximation predictions, Equations (40), (66), (69) and (70). According to Equations (40) and (66), for the values of the governing parameters considered here, the onsets of subcritical and supercritical motions are given by $R_{TC}^{sub} = 18.12$ and $R_{TC}^{sup} = 106.90$, respectively, which are independent of $\varepsilon$. As expected, the graph indicates that the onset of Hopf bifurcation, $R_{TC}^{Hopf}$, decreases sharply upon decreasing the value of the normalized porosity, $\varepsilon$. Upon decreasing the normalized porosity further, it is seen that the onset of Hopf bifurcation tends toward $R_{TC}^{Hopf} \approx R_{TC}^{sub} = 18.12$ as $\varepsilon \to 0.21$. Upon decreasing further the value of $\varepsilon$, it is observed that, in the range $0 \leq \varepsilon \leq 0.21$ there exists a threshold for the onset of the overstable regime, delineated by the hatched area. Furthermore, the overstable regime is shown to exist up to a critical Rayleigh number, $R_{TC}^{osc}$, at which the transition from the oscillatory to direct mode convection occurs. It is also observed that, for $\varepsilon = 0.21$, the condition for a codimension-2 point is reached $R_{TC}^{over} = R_{TC}^{osc} = R_{TC}^{sup} = 106.90$. Furthermore, the results presented for $\varepsilon = 0.21$ indicate that the condition $R_{TC}^{sub} = R_{TC}^{Hopf}$ is reached, as well.

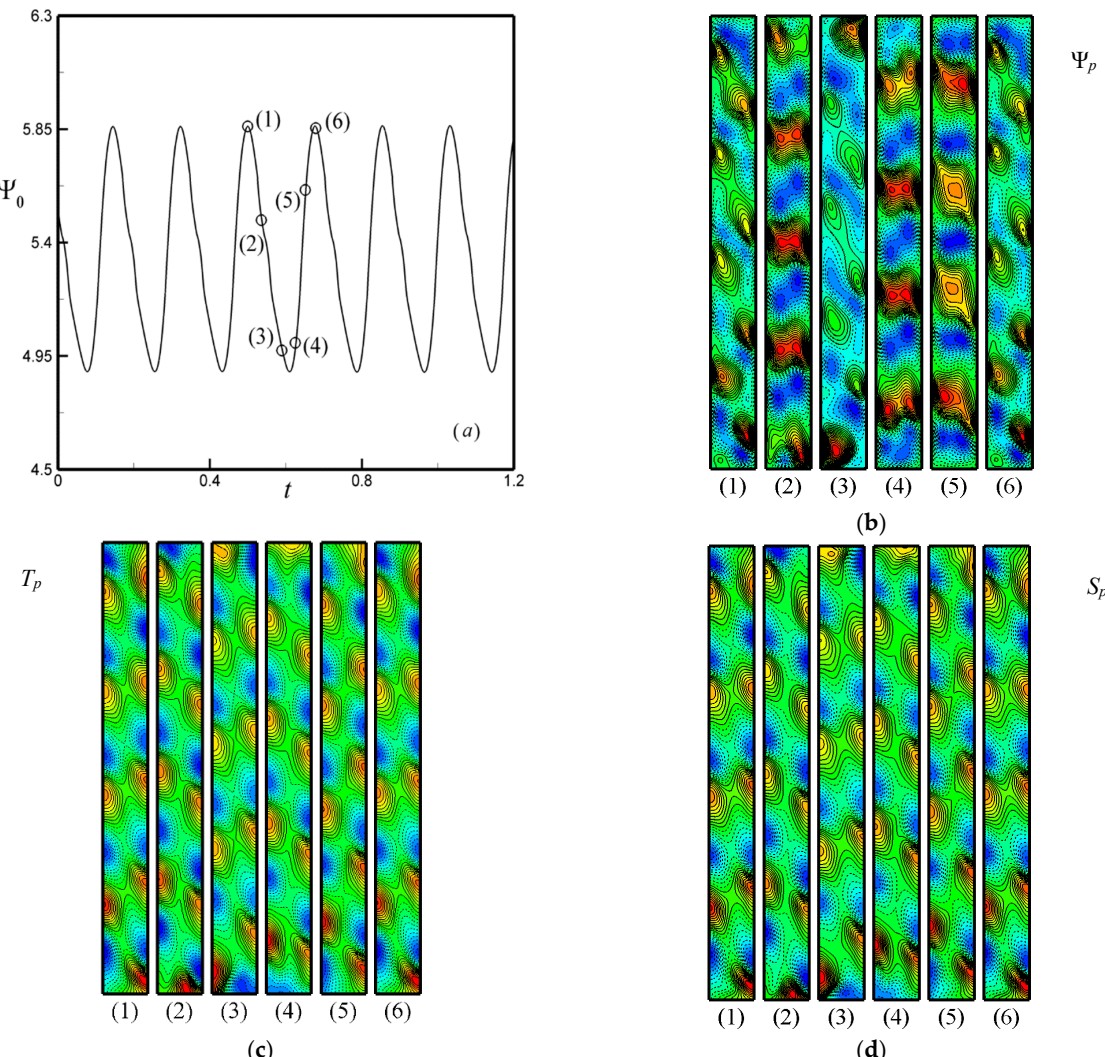

**Figure 16.** (**a**) Time evolution of the stream function, $\Psi_0$, as predicted by the numerical solution, and (**b–d**) snapshots of the perturbations of the stream function, $\Psi_p$, temperature, $T_p$ and concentration, $S_p$, for $A = 10$, $R_T = 910$, $Da_e = 10^{-4}$, $Le = 2$, $D_u = S_r = 0.1$, $\xi = 0$, $\varepsilon = 1$ and $\tau = 0.1775$.

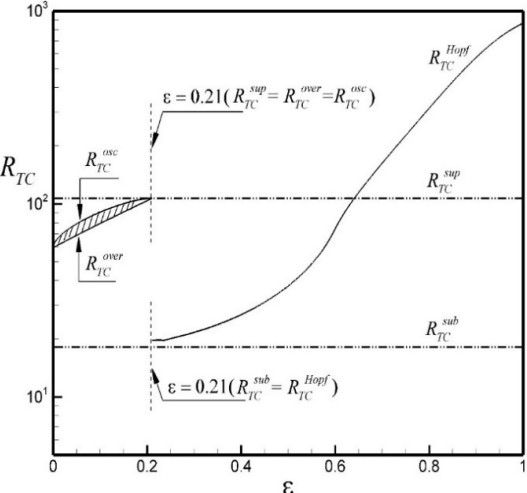

**Figure 17.** Stability diagram as a function of the normalized porosity, $\varepsilon$, for $Da_e = 10^{-4}$, $Le = 2$ and $D_u = S_r = 0.1$.

## 7. Conclusions

The combined Soret and Dufour effects on double diffusive convection in a vertical porous layer filled by a binary mixture was investigated analytically and numerically using the Darcy–Brinkman model. The particular situation where the buoyancy ratio was given by $N = -a_T/a_S$ was considered. The governing parameters of the problem were the Rayleigh number, $R_T$, effective Darcy number, $Da_e$, buoyancy ratio, $N$, Lewis number, $Le$, Soret parameter, $S_r$, Dufour number, $D_u$, normalized porosity of the porous medium, $\varepsilon$, porous medium acceleration coefficient, $\xi$, and the aspect ratio of the cavity, $A$. The influence of the Soret and Dufour effects on the strength of convective motion, heat and mass transfer rates, and on the onset of various convective modes was investigated. The main conclusions of the current study are itemized below:

1. For the large aspect ratio of the enclosure, an excellent agreement was obtained between the resulting steady state solution predicted by the asymptotic analytical theory and the numerical solution of the full governing equations. For given values of the governing parameters $Da_e$, $Le$, $R_T$, $\varepsilon$, $S_r$ and $D_u$, the strength of the convection intensity, $\Psi_0$, increased with the increase in the Rayleigh number, $R_T$ and the Dufour number, $D_u$, for $S_r < 0$, while, it decreased when $S_r > 0$. Thus, when the Rayleigh number was very large, both heat and mass transfer rates tended asymptotically toward a constant value $Nu = Sh \to 4.88$, independently of the value of both the Soret and Dufour parameters.

2. The linear stability analysis of the diffusive state was performed. For an infinite layer, the critical Rayleigh numbers characterizing the onset of stationary $\left(R_{TC}^{sup}\right)$ and oscillatory $\left(R_{TC}^{over}\right)$ convection was determined as a function of $Le$, $\varepsilon$, $S_r$ and $D_u$. The nonlinear stability analysis of the parallel flow solution indicated the existence of a subcritical Rayleigh number $\left(R_{TC}^{sub}\right)$ for the onset of finite amplitude convection, which was a function of $Le$, $S_r$ and $D_u$. The subcritical bifurcation was found to occur well below the thresholds of stationary convection. Additionally, a linear stability analysis of the steady convective solution was performed to determine the threshold of Hopf bifurcation and the results were corroborated by the numerical solution of the full governing equations.

3. The porous medium normalized porosity increase was found to have a stabilizing effect and delayed the onset of Hopf bifurcation. The Dufour parameter had a destabilizing effect, where it quickened the occurrence of the onset of the subcritical and supercritical convection and Hopf bifurcation. The Soret parameter had both stabilizing and destabilizing effects according to its value; it delayed the appearance of the subcritical and supercritical convection and Hopf bifurcation (stabilizing effect) when $S_r < 0.5$, while it reduced all the thresholds (destabilizing effect) when $S_r > 0.5$.

4. The acceleration parameter had a stabilizing effect on the convective flow and delayed transition towards oscillatory convective state.

**Author Contributions:** Conceptualization, A.B., M.M. and S.B.; methodology, A.B., M.M. and R.R.; software, A.B., M.M. and R.R.; validation, all authors; writing—original draft preparation, A.B., R.R., M.M. and S.B.; writing—review and editing, all authors. All authors have read and agreed to the published version of the manuscript.

**Funding:** This research received no external funding.

**Data Availability Statement:** The data that support the findings of this study are available from the corresponding author upon reasonable request.

**Conflicts of Interest:** The authors declare no conflict of interest.

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
