# Peer review of "Linear and Nonlinear Stability Analyses of Double-Diffusive Convection in a Vertical Brinkman Porous Enclosure under Soret and Dufour Effects"

_fluids, doi:10.3390/fluids6080292_

Round 1
Reviewer 1 Report
Linear stability theory is applied to investigate steady and oscillatory onsets. A numerical solution is used to simulate the finite amplitude onset. Double diffusion in a porous medium with Soret and Dufour effects is a complicated phenomenon. The authors did a very good job of presenting the methodologies. The stability maps were clear. I do wish the authors presented more cases of different values of D_u, S_r, ξ, and ε.
The aspect ratio was set to 10. The authors did mention that for the parameters considered in this investigation, A > 6 is large enough. However, I am a little skeptical about that. From Figures 13 and 14, I believe there are 3 or 4 periodical cells. The finite height acts as a fliter and only allows integer and/or integer +1/2 number of cells. This means the wave number cannot vary continuously. Consequently, the authors may not be simulating the most critical mode as compared to the infinitely tall channel. I am not absolutely convinced that A = 10 can represent the infinitely tall channel. The authors should clarify that.
Author Response
We would like to take this opportunity to thank you for the effort and expertise that you contribute to reviewing, for the careful reading of the manuscript and the constructive remarks., without which it would be impossible to maintain the high standards of peer-reviewed journals.
All of these comments were very helpful for revising and improving our paper. We have examined the comments and suggestions carefully and have made corresponding corrections that we hope will be satisfactory to you.
Please find below a detailed point-to-point response to all comments (reviewer comments in black, our replies in blue).
We have included a “track changes” document of the manuscript and highlighted the correction and the inserted information.

Reviewer 2 Report
The file is attached.

Author Response

(The authors gave the same response as above.)

Round 2
Reviewer 2 Report
The paper deals with linear and nonlinear stability analyses of double-diffusive convection in a vertical Brinkman porous enclosure under Soret and Dufour effects. The subject of paper will fits the scope of Fluids. The paper contains new results and could be recommended for publication in Fluids after revision taking into account the following comments:
- Justification of Eqs. (1)-(5) is needed, particularly please explain why in the left hand side time derivative is taken into account in the momentum equation and non-linear term not and why in the Laplace operator only second derivatives on one coordinate are presented.
- Is the grid 200x300 is appropriate for the rectangular cavity of aspect ratio 10?
- In most of works the Dufour effect is neglected. To justify the impoprtance of its accounting for please comment to what real fluid or class of fluids selected values of the Lewis number, the Soret parameter and the Dufour parameter could correspond.
- What could be dimensional values of the parameters to be realized for experimental observation of new phenomena found in the calculations?
- Comparison of the results with the theoretical and experimental works of other authors in limit cases would be useful.
Author Response
We would like to take this opportunity to thank you for the effort and expertise that you contributed to review the paper and for the careful reading of the manuscript and the constructive remarks. We have examined the comments and suggestions carefully and have made corrections accordingly.
Please find below a detailed point-to-point response to all your comments (reviewer comments in black, our replies in blue).
We have included a “track changes” document of the manuscript and highlighted the corrections and the necessary modifications.
